# MODEL-BASED OFFLINE PLANNING

**Arthur Argenson**
`aarg@google.com`
Google Research

**Gabriel Dulac-Arnold**
`dulacarnold@google.com`
Google Research

## ABSTRACT

Offline learning is a key part of making reinforcement learning (RL) useable in real systems. Offline RL looks at scenarios where there is data from a system's operation, but no direct access to the system when learning a policy. Recent work on training RL policies from offline data has shown results both with model-free policies learned directly from the data, or with planning on top of learnt models of the data. Model-free policies tend to be more performant, but are more opaque, harder to command externally, and less easy to integrate into larger systems. We propose an offline learner that generates a model that can be used to control the system directly through planning. This allows us to have easily controllable policies directly from data, without ever interacting with the system. We show the performance of our algorithm, Model-Based Offline Planning (`MBOP`) on a series of robotics-inspired tasks, and demonstrate its ability to leverage planning to respect environmental constraints. We are able to find near-optimal polices for certain simulated systems from as little as 50 seconds of real-time system interaction, and create zero-shot goal-conditioned policies on a series of environments.

## 1 INTRODUCTION

Learnt policies for robotic and industrial systems have the potential to both increase existing systems' efficiency & robustness, as well as open possibilities for systems previously considered too complex to control. Learnt policies also afford the possibility for non-experts to program controllers for systems that would currently require weeks of specialized work. Currently, however, most approaches for learning controllers require significant interactive time with a system to be able to converge to a performant policy. This is often either undesirable or impossible due to operating cost, safety issues, or system availability. Fortunately, many systems are designed to log sufficient data about their state and control choices to create a dataset of operator commands and resulting system states. In these cases, controllers could be learned offline, using algorithms that produce a good controller using only these logs, without ever interacting with the system. In this paper we propose such an algorithm, which we call Model-Based Offline Planning (`MBOP`), which is able to learn policies directly from logs of a semi-performant controller without interacting with the corresponding environment. It is able to leverage these logs to generate a more performant policy than the one used to generate the logs, which can subsequently be goal-conditioned or constrained dynamically during system operation.

Learning from logs of a system is often called 'Offline Reinforcement Learning' (Wu et al., 2019; Peng et al., 2019; Fujimoto et al., 2019; Wang et al., 2020) and both model-free (Wu et al., 2019; Wang et al., 2020; Fujimoto et al., 2019; Peng et al., 2019) and model-based (Yu et al., 2020; Kidambi et al., 2020) approaches have been proposed to learn policies in this setting. Current model-based approaches, MOPO (Yu et al., 2020) and MoREL (Kidambi et al., 2020), learn a model to train a model-free policy in a Dyna-like (Sutton & Barto, 2018) manner. Our proposed approach, `MBOP`, is a model-based approach that leverages Model-Predictive Control (MPC) (Rault et al., 1978) and extends the MPPI (Williams et al., 2017b) trajectory optimizer to provide a goal or reward-conditioned policy using real-time planning. It combines three main elements: a learnt world model, a learnt behavior-cloning policy, and a learnt fixed-horizon value-function.

`MBOP`'s key advantages are its data-efficiency and adaptability. `MBOP` is able to learn policies that perform better than the demonstration data from as little as 100 seconds of simulated system time (equivalent to 5000 steps). A single trained `MBOP` policy can be conditioned with a reward function,

a goal state, as well as state-based constraints, all of which can be non-stationary, allowing for easy control by a human operator or a hierarchical system. Given these two key advantages, we believe it to be a good candidate for real-world use in control systems with offline data.

We contextualize `MBOP` relative to existing work in Section 2, and describe `MBOP` in Section 3. In Section 4.2, we demonstrate `MBOP`'s performance on standard benchmark performance tasks for offline RL, and in Section 4.3 we demonstrate `MBOP`'s performance in zero-shot adaptation to varying task goals and constraints. In Section 4.4 we perform an ablation analysis and consider combined contributions of `MBOP`'s various elements.

## 2    RELATED WORKS

Model-Based approaches with neural networks have shown promising results in recent years. Guided Policy Search (Levine & Koltun, 2013) leverages differential dynamic programming as a trajectory optimizer on locally linear models, and caches the resulting piece-wise policy in a neural network. Williams et al. (2017b) show that a simple model-based controller can quickly learn to drive a vehicle on a dirt track, the BADGR robot (Kahn et al., 2020) also uses Model-Predictive Path Integral (MPPI) (Williams et al., 2017a) with a learned model to learn to navigate to novel locations, Yang et al. (2020) show good results learning legged locomotion policies using MPC with learned models, and (Ebert et al., 2018) demonstrate flexible robot arm controllers leveraging learned models with image-based goals. Silver et al. (2016) have shown the power of additional explicit planning in various board games including Go. More recently planning-based algorithms such as PlaNet (Hafner et al., 2019b) have shown strong results in pixel-based continuous control tasks by leveraging latent variational RNNs. Simpler approaches such as PDDM (Nagabandi et al., 2020) or PETS (Chua et al., 2018) have shown good results using full state information both in simulation and on real robots. `MBOP` is strongly influenced by PDDM (Nagabandi et al., 2020) (itself an extension on PETS (Chua et al., 2018)), in particular with the use of ensembles and how they are leveraged during planning. PDDM was not designed for offline use, and `MBOP` adds a value function composition as well as a policy prior during planning to increase data efficiency and strengthen the set of priors for offline learning. It leverages the same trajectory re-weighting approach used in PDDM and takes advantage of its beta-mixture of the $T$ trajectory buffer.

Both MoREL (Kidambi et al., 2020) and MOPO (Yu et al., 2020) leverage model-based approaches for offline learning. This is similar to approaches used in MBPO (Janner et al., 2019) and DREAMER (Hafner et al., 2019a), both of which leverage a learnt model to learn a model-free controller. MoREL and MOPO, however, due to their offline nature, train their model-free learner by using a surrogate MDP which penalizes for underlying model uncertainty. They do not use the models for direct planning on the problem, thus making the final policy task-specific. MOPO demonstrate the ability of their algorithm to alter the reward function and re-train a new policy according to this reward, but cannot leverage the final policy to dynamically adapt to an arbitrary goal or constrained objective. Matsushima et al. (2020) use a model-based policy for deployment efficient RL. Their use case is a mix between offline and online RL, where they consider that there is a limited number of deployments. They share a similarity in the sense that they also use a behavior-cloning policy $\pi_\beta$ to guide trajectories in a learned ensemble model, but perform policy improvement steps on a parametrized policy initialized from $\pi_\beta$ using a behavior-regularized objective function. Similarly to MoREL and MOPO their approach learns a parameterized policy for acting in the real system.

The use of a value function to extend the planning horizon of a planning-based policy has been previously proposed by Lowrey et al. (2018) with the POLO algorithm. POLO uses a ground-truth model (e.g. physics simulator) with MPPI/MPC for trajectory optimization. POLO additionally learns an approximate value-function through interaction with the environment which is then appended to optimized trajectories to improve return estimation. Aside from the fact that `MBOP` uses an entirely approximate & learned model, it uses a similar idea but with a fixed-horizon value function to avoid bootstrapping, and separate heads of the ensemble during trajectory optimization. BC-trained policies as sampling priors have been looked at by POPLIN (Wang & Ba, 2019). POPLIN does not use value bootstrapping, and re-samples an ensemble head at each timestep during rollouts, which likely provides less consistent variations in simulated plans. They show strong results relative to a series of model-based and model-free approaches, but do not manage to perform on the Gym Walker envi-

ronment. Additionally, they are overall much less data efficient than MBOP and do not demonstrate performance in the offline setting.

Task-time adaptation using model-based approaches has been considered previously in the model-based literature. Lu et al. (2019) look at mixing model-free and model-based approaches using notions of uncertainty to allow for adaptive controllers for non-stationary problems. Rajeswaran et al. (2020) use a game-theoretic framework to describe two adaptive learners that are both more sample efficient than common MBRL algorithms, as well as being more robust to non-stationary goals and system dynamics. MBOP is able to perform zero-shot adaptation to non-stationary goals and constraints, but does not provide a mechanism for dealing with non-stationary dynamics. If brought into the on-line settings, approaches from these algorithms such as concentrating on recent data, could however be leveraged to allow for this.

Previous approaches all look at various elements present in MBOP but none consider the full combination of a BC prior on the trajectory optimizer with a value-function initialization, especially in the case of full offline learning. Along with this high-level design, many implementation details such as consistent ensemble sampling during rollouts, or averaging returns over ensemble heads, appear to be important for a stable controller from our experience.

## 3 MODEL-BASED OFFLINE PLANNING

Our proposed algorithm, MBOP (Model-Based Offline Planning), is a model-based RL algorithm able to produce performant policies entirely from logs of a less-performant policy, without ever interacting with the actual environment. MBOP learns a world model and leverages a particle-based trajectory optimizer and model-predictive control (MPC) to produce a control action conditioned on the current state. It can be seen as an extension of PDDM (Nagabandi et al., 2020), with a behavior-cloned policy used as a prior on action sampling, and a fixed-horizon value function used to extend the planning horizon.

In this following sections, we introduce the Markov Decision Process (MDP) formalism, briefly explain planning-based approaches, discuss offline learning, and then introduce the elements of MBOP before describing the algorithm in full.

### 3.1 MARKOV DECISION PROCESS

Let us model our tasks as a Markov Decision Process (MDP), which can be defined as a tuple $(\mathcal{S}, \mathcal{A}, p, r, \gamma)$, where an agent is in a state $s_t \in \mathcal{S}$ and takes an action $a_t \in \mathcal{A}$ at timestep $t$. When in state $s_t$ and taking an action $a_t$, an agent will arrive in a new state $s_{t+1}$ with probability $p(s_{t+1}|s_t, a_t)$, and receive a reward $r(s_t, a_t, s_{t+1})$. The cumulative reward over a full episode is called the return $R$ and can be truncated to a specific horizon as $R_H$. Generally reinforcement learning and control aim to provide an optimal policy function $\pi^s : \mathcal{S} \to \mathcal{A}$ which will provide an action $a_t$ in state $s_t$ which will lead to the highest long-term *return*: $\pi^*(s_t) = \arg\max_{a \in \mathcal{A}} \sum_{t=1}^{\infty} \gamma^t r(s_t, \pi^*(s_t))$, where $\gamma$ is a time-wise discounting factor that we fix to $\gamma = 1$, and therefore only consider finite-horizon returns.

### 3.2 PLANNING WITH LEARNED MODELS

A large body of the contemporary work with MDPs involves Reinforcement Learning (RL) Sutton & Barto (2018) with model-free policies Mnih et al. (2015); Lillicrap et al. (2015); Schulman et al. (2017); Abdolmaleki et al. (2018). These approaches learn some form of policy network which provides its approximation of the best action $a_t$ for a given state $s_t$ often as a single forward-pass of the network. MBOP and other model-based approaches Deisenroth & Rasmussen (2011); Chua et al. (2018); Williams et al. (2017b); Hafner et al. (2019b); Lowrey et al. (2018); Nagabandi et al. (2020) are very different. They learn an approximate model of their *environment* and then use a planning algorithm to find a high-return trajectory through this model, which is then applied to the environment [1]. This is interesting because the final policy can be more easily adapted to new

---

[1]This approach is often called Model-Based Reinforcement Learning (MBRL) in the literature, but we chose to talk more generally about planning with learned models as the presence of a reward is not fundamentally necessary and the notion of *reinforcement* is much less present.

tasks, be made to respect constraints, or offer some level of explainability. When bringing learned controllers to industrial systems, many of these aspects are highly desireable, even to the expense of raw performance.

### 3.3 OFFLINE LEARNING

Most previous work in both reinforcement learning and planning with learned models has assumed repeated interactions with the target environment. This assumption allows the system to gather increased data along trajectories that are more likely, and more importantly to provides counterfactuals, able to contradict prediction errors in the learned policy, which is fundamental to policy improvement. In the case of offline learning, we consider that the environment is *not* available during the learning phase, but rather that we are given a dataset $\mathcal{D}$ of interactions with the environment, representing a series of timestep tuples $(s_t, a_t, r_t, s_{t+1})$. The goal is to provide a performant policy $\pi$ given this particular dataset $\mathcal{D}$. Existing RL algorithms do not easily port over to the offline learning setup, for a varied set of reasons well-covered in Levine et al. (2020). In our work, we use the real environment to benchmark the performance of the produced policy. It is important to point out that oftentimes there is nevertheless a need to evaluate the performance of a given policy $\pi$ *without* providing access to the final system, which is the concern of Off Policy Evaluation (OPE) Precup (2000); Nachum et al. (2019) and Offline Hyperparameter Selection(OHS) Paine et al. (2020) which are outside the scope of our contribution.

### 3.4 LEARNING DYNAMICS, ACTION PRIORS, AND VALUES

`MBOP` uses three parameterized function approximators for its planning algorithm. These are:

1. $f_m : \mathcal{S} \times \mathcal{A} \rightarrow \mathcal{S} \times \mathbb{R}$, a single-timestep model of environment dynamics such that $(\hat{r}_t, \hat{s}_{t+1}) = f_m(s_t, a_t)$. This is the model used by the planning algorithm to roll out potential action trajectories. We will use $f_m(s_t, a_t)_s$ to denote the state prediction and $f_m(s_t, a_t)_r$ for the reward prediction.

2. $f_b : \mathcal{S} \times \mathcal{A} \rightarrow \mathcal{A}$, a behavior-cloned policy network which produces $a_t = f_b(s_t, a_{t-1})$, and is used by the planning algorithm as a prior to guide trajectory sampling.

3. $f_R : \mathcal{S} \times \mathcal{A} \rightarrow \mathbb{R}$ is a truncated value function, which provides the expected return over a fixed horizon $R_H$ of taking a specific action $a$ in a state $s$, as $\hat{R}_H = f_R(s_t, a_{t-1})$.

Each one is a bootstrap ensemble (Lakshminarayanan et al., 2017) of $K$ feed-forward neural networks, thus $f_m$ is composed of $f_m^i \forall i \in [1, K]$, where each $f_m^i$ is trained with a different weight initialization but from the same dataset $\mathcal{D}$. This approach has been shown to work well empirically to stabilize planning (Nagabandi et al., 2020; Chua et al., 2018). Each of the ensemble member networks is optimized to minimize the $L_2$ loss on the predicted values in the dataset $\mathcal{D}$ in a standard supervised manner.

### 3.5 MBOP-POLICY

`MBOP` uses Model-Predictive Control (Rault et al., 1978) to provide actions for each new state as $a_t = \pi(s_t)$. MPC works by running a fixed-horizon planning algorithm at every timestep, which returns a trajectory $T$ of length $H$. MPC selects the first action from this trajectory and returns it as $a_t$. This fixed-horizon planning algorithm is effectively a black box to MPC, although in our case we have the MPC loop carry around a global trajectory buffer $T$. A high-level view of the policy loop using MPC is provided in Algorithm 1.

The `MBOP-Policy` loop is straightforward, and only needs to keep around $T$ at each timestep. MPC is well-known to be a surprisingly simple yet effective method for planning-based control. Finding a good trajectory is however more complicated, as we will see in the next section.

### 3.6 MBOP-TRAJOPT

`MBOP-Trajopt` extends ideas used by PDDM (Nagabandi et al., 2020) by adding a policy prior (provided by $f_b$) and value prediction (provided by $f_R$). The full algorithm is described in Algorithm

---

**Algorithm 1** High-Level `MBOP-Policy`

---

1: Let $\mathcal{D}$ be a dataset of $E$ episodes
2: Train $f_m, f_b, f_R$ on $\mathcal{D}$
3: Initialize planned trajectory: $T^0 = [0_0, \cdots, 0_{H-1}]$.
4: **for** $t = 1..\infty$ **do**
5:      Observe $s_t$
6:      $T^t = $ `MBOP-Trajopt`$(T^{t-1}, s_t, f_m, f_b, f_r)$          $\triangleright$ Update planned trajectory $T^t$ starting at $T_0$.
7:      $a_t = T_0^t$                              $\triangleright$ Use first action $T_0$ as $\pi(s_t)$
8: **end for**

---

**Algorithm 2** `MBOP-Trajopt`

---

1: **procedure** `MBOP-TRAJOPT`$(s, T, f_m, f_b, f_R, H, N, \sigma^2, \beta, \kappa)$
2:      Set $\mathbf{R}_N = \vec{0}_N$                        $\triangleright$ This holds our N trajectory returns.
3:      Set $\mathbf{A}_{N,H} = \vec{0}_{N,H}$          $\triangleright$ This holds our N action trajectories of length H.
4:
5:      **for** $n = 1..N$ **do**                $\triangleright$ Sample $N$ trajectories over horizon $H$.
6:          $l = n \mod K$        $\triangleright$ Use consistent ensemble head throughout trajectory.
7:          $s_1 = s, a_0 = T_0, R = 0$
8:          **for** $t = 1..H$ **do**
9:              $\epsilon \sim \mathcal{N}(0, \sigma^2)$
10:             $a_t = f_b^l(s_t, a_{t-1}) + \epsilon$         $\triangleright$ Sample current action using BC policy.
11:             $\mathbf{A}_{n,t} = (1 - \beta)a_t + \beta T_{i=\min(t, H-1)}$     $\triangleright$ Beta-mixture with previous trajectory $T$.
12:             $s_{t+1} = f_m^l(s_t, \mathbf{A}_{n,t})_s$         $\triangleright$ Sample next state from environment model.
13:             $R = R + \frac{1}{K} \sum_{i=1}^K f_m^i(s_t, \mathbf{A}_{n,t})_r$    $\triangleright$ Take average reward over all ensemble members.
14:          **end for**
15:          $\mathbf{R}_n = R + \frac{1}{K} \sum_{i=1}^K f_R^i(s_{H+1}, \mathbf{A}_{n,H})$        $\triangleright$ Append predicted return and store.
16:      **end for**
17:      $T'_t = \frac{\sum_{n=1}^N e^{\kappa \mathbf{R}_n} \mathbf{A}_{n,t+1}}{\sum_{n=1}^N e^{\kappa \mathbf{R}_n}}, \forall t \in [0, H-1]$       $\triangleright$ Generate return-weighted average trajectory.
18:      **return** $T'$
19: **end procedure**

---

2. In essence, `MBOP-Trajopt` is an iterative guided-shooting trajectory optimizer with refinement. `MBOP-Trajopt` rolls out $N$ trajectories of length $H$ using $f_m$ as an environment model. As $f_m$ is actually an ensemble with $K$ members, we denote the $l^{\text{th}}$ ensemble member as $f_m^l$. Line 6 of Alg. 2 allows the $n$th trajectory to always use the same $l$th ensemble member for both the BC policy and model steps. This use of consistent ensemble members for trajectory rollouts is inspired by PDDM. We point out that $f_m$ models return both state transitions and reward, and so we denote the state component as $f_m(s_t, a_t)_s$ and the reward component as $f_m(s_t, a_t)_r$.

The policy prior $f_b^l$ is used to sample an action which is then averaged with the corresponding action from the previous trajectory generated by `MBOP-Trajopt`. By maintaining $T$ from one MPC step to another we maintain a trajectory prior that allows us to amortize trajectory optimization over time. The $\beta$ parameter can be interpreted as a form of learning rate defining how quickly the current optimal trajectory should change with new rollout information (Wagener et al., 2019). We did not find any empirical advantage to the time-correlated noise in Nagabandi et al. (2020), instead opting for i.i.d. noise.

As opposed to the BC policy and environment model, reward model is calculated using the average over all ensemble members to calculate the expected return $R_n$ for trajectory $n$. At the end of a trajectory, we append the predicted return for the final state and action by averaging over all members of $f_R$. The decision to take an average of returns vs using the ensemble heads was also inspired by the approach used in Nagabandi et al. (2020).

Once we have a set of trajectories and their associated return, we generate an average action for timestep $t$ by re-weighting the actions of each trajectory according their exponentiated return, as in Nagabandi et al. (2020) and Williams et al. (2017b) (Alg 3, Line 17).

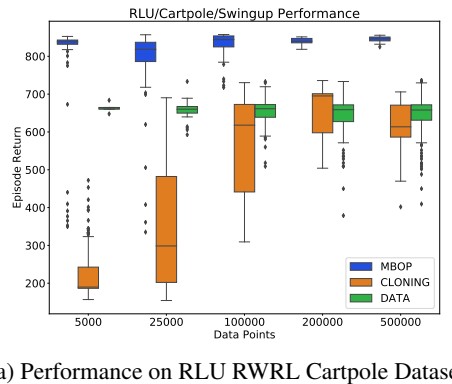

(a) Performance on RLU RWRL Cartpole Dataset

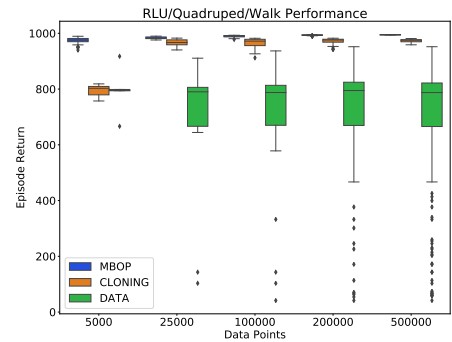

(b) Performance on RLU RWRL Quadruped Dataset

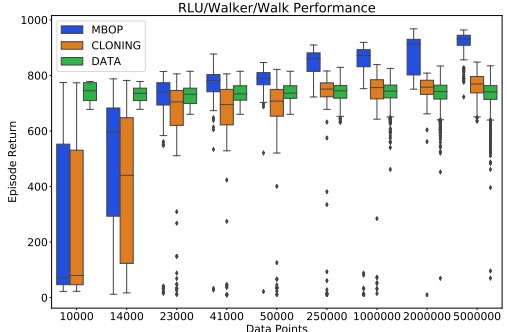

(c) Performance on RLU RWRL Walker Dataset

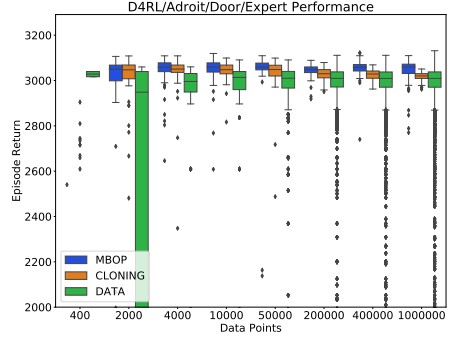

(d) Performance on D4RL Adroit Door Dataset

Figure 1: Performance of `MBOP` on various RLU and D4RL datasets. For each of the above tasks we have sub-sampled subsets of the original dataset to obtain the desired number of data points. The subsets are the same throughout the paper. The box plots describe the first quartile of the dataset, with the whiskers extending out to the full distribution, with outliers plotted individually, using the standard Seaborn (more info here).

Section 4 demonstrates how the combination of these elements makes our planning algorithm capable of generating improved trajectories over the behavior trajectories from $\mathcal{D}$, especially in low-data regimes. In higher-data regimes, variants of `MBOP` without the BC prior can also be used for goal & constraint-based control. Further work will consider the addition of goal-conditioned $f_b$ and $f_R$ to allow for more data-efficient goal and constraint-based control.

## 4 EXPERIMENTAL RESULTS

We look at two operating scenarios to demonstrate `MBOP` performance and flexibility. First we consider the standard offline settings where the evaluation environment and task are identical to the behavior policy's. We show that `MBOP` is able to perform well with very little data. We then look at `MBOP`'s ability to provide controllers that can naturally transfer to novel tasks with the same system dynamics. We use both goal-conditioned tasks (that ignore the original reward function) and constrained tasks (that require optimising for the original reward under some state constraint) to demonstrate the `MBOP`'s transfer abilities. Accompanying videos are available here: `https://youtu.be/nxGGHdZOFts`.

### 4.1 METHODOLOGY

We use standard datasets from the RL Unplugged (RLU) (Gulcehre et al., 2020) and D4RL (Fu et al., 2020) papers. For both RLU and D4RL, policies are trained from offline datasets and then evaluated on the corresponding environment. For datasets with high variance in performance, we discard episodes that are below a certain threshold for the training of $f_b$ and $f_R$. This is only done on the `Quadruped` and `Walker` tasks from RLU, and only provides a slight performance boost – performance on unfiltered data for these two tasks can be found in the Appendix's 5.6. The unfiltered data is always used for training $f_s$. We perform a grid-search to find optimal parameters for each dataset, but for most tasks these parameters are mostly uniform. The full set of parameters for each experiment can be found in the Appendix Sec. 5.2. For experiments on RLU, we generated

| Dataset type | Environment | BC (Ours) | MBOP (Ours) | MOPO | MBPO |
|---|---|---|---|---|---|
| random | halfcheetah | $0.0 \pm 0.0$ | $6.3 \pm 4.0$ | $\mathbf{31.9} \pm 2.8$ | $30.7 \pm 3.9$ |
| random | hopper | $9.0 \pm 0.2$ | $10.8 \pm 0.3$ | $\mathbf{13.3} \pm 1.6$ | $4.5 \pm 6.0$ |
| random | walker2d | $0.1 \pm 0.0$ | $8.1 \pm 5.5$ | $\mathbf{13.0} \pm 2.6$ | $8.6 \pm 8.1$ |
| medium | halfcheetah | $35.0 \pm 2.5$ | $\mathbf{44.6} \pm 0.8$ | $40.2 \pm 2.7$ | $28.3 \pm 22.7$ |
| medium | hopper | $48.1 \pm 26.2$ | $\mathbf{48.8} \pm 26.8$ | $26.5 \pm 3.7$ | $4.9 \pm 3.3$ |
| medium | walker2d | $15.4 \pm 24.7$ | $\mathbf{41.0} \pm 29.4$ | $14.0 \pm 10.1$ | $12.7 \pm 7.6$ |
| mixed | halfcheetah | $0.0 \pm 0.0$ | $42.3 \pm 0.9$ | $\mathbf{54.0} \pm 2.6$ | $47.3 \pm 12.6$ |
| mixed | hopper | $9.5 \pm 6.9$ | $12.4 \pm 5.8$ | $\mathbf{92.5} \pm 6.3$ | $49.8 \pm 30.4$ |
| mixed | walker2d | $11.5 \pm 7.3$ | $9.7 \pm 5.3$ | $\mathbf{42.7} \pm 8.3$ | $22.2 \pm 12.7$ |
| med-expert | halfcheetah | $90.8 \pm 26.9$ | $\mathbf{105.9} \pm 17.8$ | $57.9 \pm 24.8$ | $9.7 \pm 9.5$ |
| med-expert | hopper | $15 \pm 8.7$ | $55.1 \pm 44.3$ | $51.7 \pm 42.9$ | $\mathbf{56.0} \pm 34.5$ |
| med-expert | walker2d | $65.5 \pm 40.2$ | $\mathbf{70.2} \pm 36.2$ | $55.0 \pm 19.1$ | $7.6 \pm 3.7$ |

Table 1: Results for `MBOP` on D4RL tasks compare to MOPO (Yu et al., 2020) and MBPO (Janner et al., 2019), with values taken from the MOPO paper (Yu et al., 2020). As in Fu et al. (2020), we normalize the scores according to a converged SAC policy, reported in their appendix. Scores are reported averaged over 5 random seeds, with 20 episode runs per seed. $\pm$ is one standard deviation and represents variance due to seed and episode. We have inserted our BC prior as the BC baseline, and have set performance to 0.0 when it is negative. We include the performance of behavior cloning (**BC**) from the batch data for comparison. We bold the highest mean.

additional smaller datasets to increase the difficulty of the problem. On all plots we also report the performance of the behavior policy used to generate the data (directly from the episode returns in the datasets) and label it as the DATA policy. All non-standard datasets will be available publicly.

For RLU the datasets are generated using a 70% performant MPO (Abdolmaleki et al., 2018) policy on the original task, and smaller versions of the datasets are a fixed set of randomly sampled contiguous episodes (Dulac-Arnold et al., 2020; Gulcehre et al., 2020). D4RL has 4 behavior policies, ranging from random behavior to expert demonstrations, and are fully described in Fu et al. (2020). On all datasets, training is performed on 90% of data and 10% is used for validation.

## 4.2 PERFORMANCE ON RL-UNPLUGGED & D4RL

For experiments on RLU we consider the unperturbed RWRL `cartpole-swingup`, `walker` and `quadruped` tasks (Tassa et al., 2018; Dulac-Arnold et al., 2020). For D4RL we consider the `halfcheetah`, `hopper`, `walker2d` and `Adroit` tasks (Brockman et al., 2016; Rajeswaran et al., 2017). Results for the RLU tasks as well as `Adroit` are presented in Figure 1. On the remaining D4RL tasks, results are compared to those presented by MOPO Yu et al. (2020) in Table 1 for four different data regimes (`medium`, `medium-expert`, `medium-replay`, `random`). For all experiments we report `MBOP` performance as well as the performance of a behavior cloning (BC) policy. The BC policy is simply the policy prior $f_b$, with the control action as the average ensemble output. We use this baseline to demonstrate the advantages brought about by planning beyond simple cloning.

For the RLU datasets (Fig. 1), we observe that `MBOP` is able to find a near-optimal policy on most dataset sizes in `Cartpole` and `Quadruped` with as little as 5000 steps, which corresponds to 5 episodes, or approximately 50 seconds on `Cartpole` and 100 seconds on `Quadruped`. On the `Walker` datasets `MBOP` requires 23 episodes (approx. 10 minutes) before it finds a reasonable policy, and with sufficient data converges to a score of 900 which is near optimal. On most tasks, `MBOP` is able to generate a policy significantly better than the behavior data as well as the the BC prior.

For the `Adroit` task, we show that `MBOP` is able to outperform the behavior policy after training on a dataset of 50k data points generated by an expert policy (Fig. 1d). For other D4RL datasets, we compare to the performance of MOPO (Yu et al., 2020). We show that on the `medium` and `medium-expert` data regimes `MBOP` outperforms MOPO, sometimes significantly. However on higher-variance datasets such as `random` and `mixed` `MBOP` is not as performant. This is likely due to the reliance on policy-conditioned priors, which we hope to render more flexible in future work (for instance using multi-modal stochastic models). There are nevertheless many tasks where a human operator is running a systems in a relatively consistent yet sub-optimal manner, and one

may want to either replicate or improve upon the operator's control policy. In such scenarios, MBOP would likely be able to not only replicate but improve upon the operator's control strategy.

### 4.3 Zero-Shot Task Adaptation

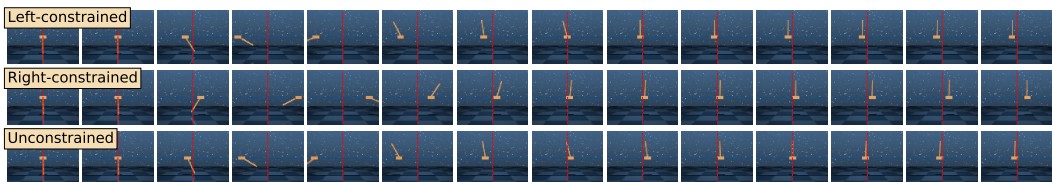

(a) Visualized trajectories for constrained Cartpole.

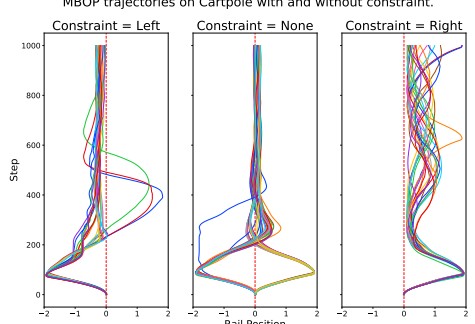

(b) RLU Cartpole trajectories per constraint type

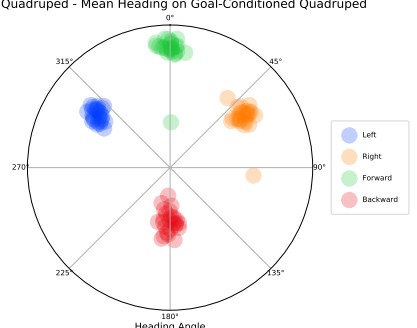

(c) RLU Goal-Directed Quadruped

Figure 2: The above figures describe performance of MBOP on constrained & goal-conditioned tasks. Fig. 2a illustrates a sequences of frames from the RLU Cartpole task with constrained and unconstrained MBOP controllers. In the constrained cases MBOP prevents the cart from crossing the middle of the rail (dotted red line) and contains it to one side. Fig. 2b displays cart trajectories for constrained and unconstrained versions of the same controller. MBOP can maintain a performant policy (above 750) while respecting these constraints. Fig. 2c displays goal-conditioned performance on the RLU Quadruped. We ignore the original reward function and optimize directly for trajectories that maximize a particular velocity vector. Although influence from $f_B$ and $f_R$ biases the controller to maintain forward direction, we can still exert significant goal-directed influence on the policy.

One of the main advantages of using planning-based methods in the offline scenario is that they are easy to adapt to new objective functions. In the case of MBOP these would be novel objectives different from those optimized by the behavior policy that generates the offline data. We can easily take these new objectives into account by computing a secondary objective return as follows: $\mathbf{R}'_n = \sum_t f_{obj}(s_t)$ where $f_{obj}$ is a user-provided function that computes a scalar objective reward given a state. We can then adapt the trajectory update rule to take into account the secondary objective:

$$T_t = \frac{\sum_{n=1}^{N} e^{\kappa \mathbf{R}_n + \kappa_{\mathrm{obj}} \mathbf{R}'_n} \mathbf{A}_{n,t}}{\sum_{n=1}^{N} e^{\kappa \mathbf{R}_n + \kappa_{obj} \mathbf{R}'_n}}, \forall t \in [1, H].$$

To demonstrate this, we run MBOP on two types of modified objectives: goal-conditioned control, and constrained control. In goal-conditioned control, we ignore the original reward function ($\kappa = 0$) and define a new goal (such as a velocity vector) and optimize trajectories relative to that goal. In constrained operation, we add a state-based constraint which we penalize during planning, while maintaining the original objective and find a reasonable combination of $\kappa$ and $\kappa_{\mathrm{obj}}$.

We define three tasks: position-constrained Cartpole, where we penalize the cart's position to encourage it to stay either on the right or left side of the track; heading-conditioned Quadruped, where we provide a target heading to the policy (Forward, Backwards, Right & Left); and finally height-constrained Walker, where we penalize the policy for bringing the torso height above a certain threshold. Results on Cartpole & Quadruped are presented in Figure 2.

We show that MBOP successfully integrates constraints that were not initially in the dataset and is able to perform well on objectives that are different from the objective of the behavior policy.

`Walker` performs similarly, obtaining nearly 80% constraint satisfaction while maintaining a reward of 730. More analysis is available in the Appendix Sec. 5.5.

## 4.4 ALGORITHMIC INVESTIGATIONS

**Ablations** To better understand the benefits of `MBOP`'s various elements, we perform three ablations: `MBOP-NOPP` which replaces $f_b$ with a Gaussian prior, `MBOP-NOVF` which removes $f_R$'s estimated returns, and `PDDM` which removes both, thus recovering the `PDDM` controller. We show performance of these four ablations on the Walker dataset in Fig. 3a. A full set of ablations is available in the appendix Figures 4 & 5. Overall we see that the full combination of BC prior, value function and environment model are important for optimal performance. We also see that the PDDM approach is generally below either of the `MBOP-NOPP` and `MBOP-NOVF` ablations. Finally, we note that the BC prior when used alone can perform well on certain environments, but on others it stagnates at behavior policy's performance.

**Execution Speed** A frequent concern with planning-based methods is their slower response time prohibiting practical use. We calculate the average control frequency of `MBOP` on the RLU Walker task using a single Intel(R) Xeon(R) W-2135 CPU @ 3.70GHz core and a Nvidia 1080TI and find that `MBOP` can operate at frequencies ranging from 106 Hz for $h = 4$ to 40 Hz for a $h = 40$, with BC operating at 362 Hz. Additional values are presented in Appendix Sec. 5.4.

**Hyperparameter Stability**

We perform a grid sweep over the $\kappa$ (trajectory re-weighting) and $H$ (planning horizon) on the three RLU environments and visualize the effects on return in Fig. 3b. We observe that overall `MBOP` maintains consistent performance scores for wide ranges of hyperparameter values, only really degrading near extreme values. Additional analysis is present in the Appendix's Section 5.5.

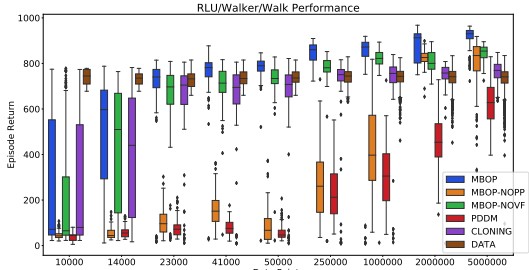

(a) `MBOP` ablations' performance on RLU Walker Dataset. We observe that `MBOP` is consistently more performant than its ablations.

## 5 CONCLUSION

Planning-based methods provide significantly more flexibility for external systems to interact with the learned controller. Bringing them into the offline data regime opens the door to their use on more real-world systems for which online training is not an option.

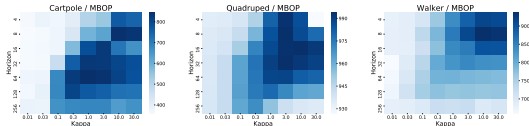

(b) `MBOP` sensitivity to Kappa ($\kappa$) and Horizon ($H$).

`MBOP` provides an easy to implement, data-efficient, stable, and flexible algorithm for policy generation. It is easy to implement because the learning components are simple supervised learners, it is data-efficient thanks to its use of multiple complementary estimators, and it is flexible due to its use of on-line planning which allows it to dynamically react to changing goals, costs and environmental constraints.

We show that `MBOP` can perform competitively in various data regimes, and can provide easily adaptable policies for more complex goal-conditioned or constrained tasks, even if the original data does not provide prior experience. Although `MBOP`'s performance is degraded when offline data is multi-modal or downright random, we believe there are a large number of scenarios where the current operating policy (be it human or automated) is reasonably consistent, but could benefit from being automated and improved upon. In these scenarios we believe that `MBOP` could be readily applicable. Future work intends to ameliorate performance by investigating the use of goal-conditioned policy priors and value estimates, as well as looking at effective ways to perform offline model selection and evaluation. We sincerely hope that `MBOP` can be useful as an out-of-the-box algorithm for learning stable and configurable control policies for real systems.

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

## APPENDIX

### 5.1 MBOP PERTINENCE TO ROBOTICS

MBOP provides a general model-based approach for offline learning. We have considered only physics-bound tasks in this paper as the underlying methods (MPC, MPPI) are known to work well on real systems (Nagabandi et al., 2020; Williams et al., 2015; Kahn et al., 2020). Although this paper does not implement MBOP on actual robots, this is upcoming work, and we believe that by having shown MBOP's performance over 6 different environments (cartpole, walker, quadruped, Adroit, halfcheetah, hopper) involving under-actuated control, locomotion, and manipulation, MBOP's potential for applicability on a real systems is promising. More specifically, we believe MBOP provides a couple key contributions specifically interesting to the robotics community:

- Ability to learn entirely offline without a simulator.
- Ability to constrain policy operation.
- Ability to completely rephrase the policy's goal according to an arbitrary cost function.

These aspects make MBOP a unique contribution that potentially opens a series of interesting research questions around zero-shot adaptation, leveraging behavior priors, using sub-optimal models, leveraging uncertainty, and more generally exploring the additional control opportunities provided by model-based methods that are much more difficult with model-free learnt controllers.

As mentioned above it is our intent to quickly try out MBOP on various robotic systems. If results are available by the time of CoRL 2020 they will be presented as well.

### 5.2 PERFORMANCE OF MBOP ABLATIONS AND ASSOCIATED HYPERPARAMETERS

We present mean evaluation performance and associated hyper parameters for runs of MBOP and its ablations in a set of tables. For RLU: Table 2 for Cartpole, 3 for Quadruped, 4 for Walker. For D4RL:

| # Points | Policy | Horizon | # Samples | Kappa | Sigma | Beta | Mean | 1-STD |
|---|---|---|---|---|---|---|---|---|
| 5000 | CLONING | - | - | - | - | - | 229.2 | 71.7 |
| 5000 | MBOP | 64 | 100 | 2.34 | 0.8 | 0.2 | 803.1 | 117.7 |
| 5000 | MBOP-NOPP | 128 | 100 | 0.23 | 0.8 | 0.2 | 605.8 | 223.6 |
| 5000 | MBOP-NOVF | 128 | 100 | 1.17 | 0.8 | 0.2 | 715.2 | 183.7 |
| 5000 | PDDM | 128 | 100 | 0.7 | 0.8 | 0.2 | 726.6 | 131.8 |
| 25000 | CLONING | - | - | - | - | - | 350.7 | 168.2 |
| 25000 | MBOP | 64 | 100 | 0.5 | 0.8 | 0.2 | 792.0 | 90.4 |
| 25000 | MBOP-NOPP | 64 | 100 | 2.3 | 0.1 | 0.2 | 463.6 | 284.0 |
| 25000 | MBOP-NOVF | 128 | 100 | 0.7 | 0.8 | 0.2 | 776.5 | 128.7 |
| 25000 | PDDM | 128 | 100 | 0.7 | 0.4 | 0.2 | 720.2 | 69.3 |
| 100000 | CLONING | - | - | - | - | - | 567.4 | 123.6 |
| 100000 | MBOP | 64 | 100 | 2.3 | 0.8 | 0.2 | 834.4 | 28.6 |
| 100000 | MBOP-NOPP | 64 | 100 | 1.4 | 0.2 | 0.2 | 832.1 | 51.1 |
| 100000 | MBOP-NOVF | 128 | 100 | 1.2 | 1.6 | 0.2 | 733.6 | 150.3 |
| 100000 | PDDM | 128 | 100 | 1.2 | 0.2 | 0.2 | 723.5 | 124.8 |
| 200000 | CLONING | - | - | - | - | - | 644.3 | 78.9 |
| 200000 | MBOP | 64 | 100 | 0.5 | 1.6 | 0.2 | 840.7 | 7.5 |
| 200000 | MBOP-NOPP | 64 | 100 | 2.3 | 0.2 | 0.2 | 840.6 | 12.4 |
| 200000 | MBOP-NOVF | 128 | 100 | 1.2 | 1.6 | 0.2 | 767.0 | 83.6 |
| 200000 | PDDM | 128 | 100 | 1.2 | 0.2 | 0.2 | 797.6 | 47.1 |
| 500000 | CLONING | - | - | - | - | - | 612.0 | 63.9 |
| 500000 | MBOP | 64 | 100 | 1.4 | 1.6 | 0.2 | 845.7 | 6.7 |
| 500000 | MBOP-NOPP | 64 | 100 | 2.3 | 0.2 | 0.2 | 840.6 | 13.7 |
| 500000 | MBOP-NOVF | 128 | 100 | 1.2 | 1.6 | 0.2 | 823.2 | 44.2 |
| 500000 | PDDM | 128 | 100 | 1.2 | 0.2 | 0.2 | 781.9 | 96.6 |

Table 2: RLU Cartpole Performance

| # Points | Policy | Horizon | # Samples | Kappa | Sigma | Beta | Mean | 1-STD |
|---|---|---|---|---|---|---|---|---|
| 5000 | CLONING | - | - | - | - | - | 796.0 | 17.0 |
| 5000 | MBOP | 8 | 1000 | 3.8 | 0.8 | 0.2 | 974.1 | 9.9 |
| 5000 | MBOP-NOPP | 16 | 1000 | 1.9 | 1.6 | 0.2 | 561.6 | 233.2 |
| 5000 | MBOP-NOVF | 16 | 1000 | 1.9 | 0.8 | 0.2 | 959.5 | 15.1 |
| 5000 | PDDM | 32 | 1000 | 0.9 | 1.6 | 0.2 | 569.8 | 26.9 |
| 25000 | CLONING | - | - | - | - | - | 966.5 | 10.9 |
| 25000 | MBOP | 8 | 1000 | 3.8 | 0.8 | 0.2 | 983.8 | 3.6 |
| 25000 | MBOP-NOPP | 16 | 1000 | 1.9 | 1.6 | 0.2 | 866.6 | 87.4 |
| 25000 | MBOP-NOVF | 16 | 1000 | 1.9 | 0.8 | 0.2 | 983.0 | 1.6 |
| 25000 | PDDM | 32 | 1000 | 0.9 | 1.6 | 0.2 | 728.1 | 120.7 |
| 100000 | CLONING | - | - | - | - | - | 966.5 | 15.1 |
| 100000 | MBOP | 8 | 1000 | 3.8 | 0.8 | 0.2 | 989.4 | 3.2 |
| 100000 | MBOP-NOPP | 16 | 1000 | 1.9 | 1.6 | 0.2 | 935.3 | 35.2 |
| 100000 | MBOP-NOVF | 16 | 1000 | 1.9 | 0.8 | 0.2 | 983.8 | 2.2 |
| 100000 | PDDM | 32 | 1000 | 0.9 | 1.6 | 0.2 | 967.1 | 12.5 |
| 200000 | CLONING | - | - | - | - | - | 972.9 | 8.6 |
| 200000 | MBOP | 8 | 1000 | 3.8 | 0.8 | 0.2 | 993.3 | 1.3 |
| 200000 | MBOP-NOPP | 16 | 1000 | 1.9 | 1.6 | 0.2 | 984.5 | 12.1 |
| 200000 | MBOP-NOVF | 16 | 1000 | 1.9 | 0.8 | 0.2 | 986.6 | 1.3 |
| 200000 | PDDM | 32 | 1000 | 0.9 | 1.6 | 0.2 | 946.4 | 29.7 |
| 500000 | CLONING | - | - | - | - | - | 973.1 | 5.6 |
| 500000 | MBOP | 8 | 1000 | 3.8 | 0.8 | 0.2 | 994.8 | 0.4 |
| 500000 | MBOP-NOPP | 16 | 1000 | 1.9 | 1.6 | 0.2 | 994.0 | 3.5 |
| 500000 | MBOP-NOVF | 16 | 1000 | 1.9 | 0.8 | 0.2 | 984.2 | 2.2 |
| 500000 | PDDM | 32 | 1000 | 0.9 | 1.6 | 0.2 | 965.0 | 12.0 |

Table 3: RLU-Quadruped Performance

| # Points | Policy | Horizon | # Samples | Kappa | Sigma | Beta | Mean | 1-STD |
|---|---|---|---|---|---|---|---|---|
| 10000 | CLONING | - | - | - | - | - | 244.5 | 284.6 |
| 10000 | MBOP | 8 | 100 | 3.8 | 0.2 | 0.2 | 251.2 | 280.4 |
| 10000 | MBOP-NOPP | 32 | 100 | 4.7 | 1.6 | 0.2 | 45.7 | 16.6 |
| 10000 | MBOP-NOVF | 4 | 100 | 7.5 | 0.1 | 0.2 | 225.3 | 275.8 |
| 10000 | PDDM | 8 | 100 | 18.8 | 0.8 | 0.2 | 37.0 | 16.8 |
| 14000 | CLONING | - | - | - | - | - | 402.4 | 263.0 |
| 14000 | MBOP | 4 | 100 | 37.5 | 0.1 | 0.2 | 489.7 | 250.3 |
| 14000 | MBOP-NOPP | 32 | 100 | 2.8 | 1.6 | 0.2 | 53.1 | 24.0 |
| 14000 | MBOP-NOVF | 4 | 100 | 37.5 | 0.1 | 0.2 | 424.3 | 266.2 |
| 14000 | PDDM | 32 | 100 | 4.7 | 1.6 | 0.2 | 57.4 | 23.2 |
| 23000 | CLONING | - | - | - | - | - | 616.7 | 224.4 |
| 23000 | MBOP | 16 | 100 | 1.9 | 0.2 | 0.2 | 679.0 | 200.2 |
| 23000 | MBOP-NOPP | 8 | 100 | 18.8 | 0.8 | 0.2 | 103.0 | 56.2 |
| 23000 | MBOP-NOVF | 8 | 100 | 18.8 | 0.1 | 0.2 | 617.7 | 220.8 |
| 23000 | PDDM | 32 | 100 | 4.7 | 1.6 | 0.2 | 77.6 | 40.4 |
| 41000 | CLONING | - | - | - | - | - | 638.0 | 200.2 |
| 41000 | MBOP | 8 | 100 | 3.8 | 0.2 | 0.2 | 752.0 | 118.0 |
| 41000 | MBOP-NOPP | 8 | 100 | 11.3 | 1.6 | 0.2 | 160.9 | 80.5 |
| 41000 | MBOP-NOVF | 32 | 100 | 2.8 | 0.1 | 0.2 | 700.6 | 97.7 |
| 41000 | PDDM | 32 | 100 | 4.7 | 0.8 | 0.2 | 79.5 | 32.9 |
| 50000 | CLONING | - | - | - | - | - | 615.7 | 240.6 |
| 50000 | MBOP | 4 | 100 | 7.5 | 0.4 | 0.2 | 775.0 | 87.1 |
| 50000 | MBOP-NOPP | 4 | 100 | 22.5 | 1.6 | 0.2 | 87.4 | 78.2 |
| 50000 | MBOP-NOVF | 16 | 100 | 9.4 | 0.2 | 0.2 | 723.2 | 103.1 |
| 50000 | PDDM | 32 | 100 | 2.8 | 1.6 | 0.2 | 59.4 | 33.6 |
| 250000 | CLONING | - | - | - | - | - | 686.8 | 205.4 |
| 250000 | MBOP | 4 | 100 | 7.5 | 0.4 | 0.2 | 844.7 | 48.4 |
| 250000 | MBOP-NOPP | 4 | 100 | 22.5 | 1.6 | 0.2 | 269.1 | 155.9 |
| 250000 | MBOP-NOVF | 16 | 100 | 9.4 | 0.2 | 0.2 | 770.4 | 115.1 |
| 250000 | PDDM | 32 | 100 | 2.8 | 1.6 | 0.2 | 231.3 | 112.2 |
| 1000000 | CLONING | - | - | - | - | - | 701.5 | 190.9 |
| 1000000 | MBOP | 4 | 100 | 7.5 | 0.4 | 0.2 | 797.3 | 229.5 |
| 1000000 | MBOP-NOPP | 4 | 100 | 22.5 | 1.6 | 0.2 | 411.2 | 183.5 |
| 1000000 | MBOP-NOVF | 16 | 100 | 9.4 | 0.2 | 0.2 | 814.3 | 88.4 |
| 1000000 | PDDM | 32 | 100 | 2.8 | 1.6 | 0.2 | 308.2 | 140.3 |
| 2000000 | CLONING | - | - | - | - | - | 743.6 | 85.6 |
| 2000000 | MBOP | 4 | 100 | 7.5 | 0.4 | 0.2 | 872.4 | 70.2 |
| 2000000 | MBOP-NOPP | 4 | 100 | 22.5 | 1.6 | 0.2 | 823.8 | 35.5 |
| 2000000 | MBOP-NOVF | 16 | 100 | 9.4 | 0.2 | 0.2 | 807.8 | 44.2 |
| 2000000 | PDDM | 32 | 100 | 2.8 | 1.6 | 0.2 | 460.3 | 117.6 |
| 5000000 | CLONING | - | - | - | - | - | 759.9 | 48.2 |
| 5000000 | MBOP | 4 | 100 | 7.5 | 0.4 | 0.2 | 908.8 | 54.3 |
| 5000000 | MBOP-NOPP | 4 | 100 | 22.5 | 1.6 | 0.2 | 784.0 | 140.8 |
| 5000000 | MBOP-NOVF | 16 | 100 | 9.4 | 0.2 | 0.2 | 833.5 | 100.7 |
| 5000000 | PDDM | 32 | 100 | 2.8 | 1.6 | 0.2 | 620.1 | 98.7 |

Table 4: RLU-Walker Performance

| # Points | Policy | Horizon | # Samples | Kappa | Sigma | Beta | Mean | 1-STD |
|---|---|---|---|---|---|---|---|---|
| 400 | CLONING | - | - | - | - | - | 352.2 | 942.3 |
| 400 | MBOP | 32 | 100 | 0.03 | 0.05 | 0 | 22.7 | 348.5 |
| 400 | MBOP-NOPP | 16 | 200 | 0.01 | 0.05 | 0 | -54.6 | 1.0 |
| 400 | MBOP-NOVF | 4 | 500 | 0.01 | 0.05 | 0 | 202.3 | 779.7 |
| 400 | PDDM | 4 | 500 | 0.01 | 0.05 | 0.2 | -52.9 | 0.6 |
| 2000 | CLONING | - | - | - | - | - | 2889.5 | 579.7 |
| 2000 | MBOP | 8 | 100 | 0.03 | 0.05 | 0 | 2944.8 | 398.6 |
| 2000 | MBOP-NOPP | 8 | 200 | 0.01 | 0.05 | 0 | -54.1 | 0.6 |
| 2000 | MBOP-NOVF | 16 | 1000 | 0.03 | 0.1 | 0 | 2903.7 | 537.2 |
| 2000 | PDDM | 4 | 500 | 0.01 | 0.05 | 0.2 | -53.0 | 0.6 |
| 4000 | CLONING | - | - | - | - | - | 3019.1 | 180.4 |
| 4000 | MBOP | 16 | 200 | 0.01 | 0.05 | 0 | 3043.4 | 64.3 |
| 4000 | MBOP-NOPP | 64 | 200 | 0.03 | 0.4 | 0 | -61.1 | 2.6 |
| 4000 | MBOP-NOVF | 64 | 200 | 0.03 | 0.05 | 0 | 2991.9 | 302.7 |
| 4000 | PDDM | 4 | 500 | 0.03 | 0.05 | 0.2 | -52.8 | 0.6 |
| 10000 | CLONING | - | - | - | - | - | 2980.3 | 335.3 |
| 10000 | MBOP | 4 | 100 | 0.3 | 0.05 | 0 | 3026.1 | 180.7 |
| 10000 | MBOP-NOPP | 8 | 500 | 0.01 | 0.05 | 0 | -53.5 | 0.6 |
| 10000 | MBOP-NOVF | 16 | 100 | 0.03 | 0.05 | 0 | 2973.6 | 351.5 |
| 10000 | PDDM | 4 | 100 | 0.01 | 0.05 | 0.2 | -53.1 | 0.7 |
| 50000 | CLONING | - | - | - | - | - | 2984.5 | 313.4 |
| 50000 | MBOP | 4 | 1000 | 0.03 | 0.2 | 0 | 3028.2 | 197.6 |
| 50000 | MBOP-NOPP | 4 | 100 | 0.01 | 0.05 | 0 | -53.4 | 0.9 |
| 50000 | MBOP-NOVF | 64 | 100 | 0.3 | 0.05 | 0 | 3052.2 | 28.2 |
| 50000 | PDDM | 4 | 500 | 0.01 | 0.1 | 0.2 | -53.0 | 0.8 |
| 200000 | CLONING | - | - | - | - | - | 3028.0 | 26.6 |
| 200000 | MBOP | 8 | 1000 | 0.03 | 0.2 | 0 | 2967.0 | 355.0 |
| 200000 | MBOP-NOPP | 4 | 500 | 0.01 | 0.05 | 0 | -52.9 | 0.8 |
| 200000 | MBOP-NOVF | 32 | 500 | 0.3 | 0.1 | 0 | 3024.2 | 198.6 |
| 200000 | PDDM | 64 | 500 | 0.3 | 0.2 | 0.2 | -59.7 | 2.8 |
| 400000 | CLONING | - | - | - | - | - | 3025.1 | 21.0 |
| 400000 | MBOP | 16 | 100 | 0.3 | 0.1 | 0 | 3000.3 | 388.4 |
| 400000 | MBOP-NOPP | 64 | 100 | 0.03 | 0.4 | 0 | -61.4 | 2.1 |
| 400000 | MBOP-NOVF | 16 | 200 | 0.3 | 0.1 | 0 | 3019.5 | 128.9 |
| 400000 | PDDM | 4 | 1000 | 0.01 | 0.1 | 0.2 | -52.9 | 0.6 |
| 1000000 | CLONING | - | - | - | - | - | 3004.3 | 142.3 |
| 1000000 | MBOP | 16 | 100 | 0.1 | 0.2 | 0 | 2910.2 | 579.6 |
| 1000000 | MBOP-NOPP | 64 | 1000 | 0.01 | 0.4 | 0 | -60.7 | 2.0 |
| 1000000 | MBOP-NOVF | 32 | 200 | 0.3 | 0.1 | 0 | 3015.6 | 241.6 |
| 1000000 | PDDM | 16 | 100 | 0.01 | 0.4 | 0.2 | -54.6 | 1.7 |

Table 5: D4RL Door Performance

| Dataset | Policy | Horizon | # Samples | Kappa | Sigma | Beta | Mean | 1-STD |
|---|---|---|---|---|---|---|---|---|
| med-expert | CLONING | - | - | - | - | - | 11012.8 | 3259.7 |
| med-expert | MBOP | 2 | 100 | 1 | 0.2 | 0 | 12850.7 | 2160.7 |
| med-expert | MBOP-NOPP | 2 | 100 | 1 | 0.2 | 0 | -334.1 | 92.2 |
| med-expert | MBOP-NOVF | 40 | 100 | 1 | 0.2 | 0 | 7220.3 | 3450.9 |
| med-expert | PDDM | 2 | 100 | 1 | 0.2 | 0 | -165.2 | 35.8 |
| mixed | CLONING | - | - | - | - | - | -6.0 | 1.6 |
| mixed | MBOP | 4 | 100 | 3 | 0.2 | 0 | 5135.1 | 107.9 |
| mixed | MBOP-NOPP | 4 | 100 | 3 | 0.2 | 0 | -415.7 | 43.3 |
| mixed | MBOP-NOVF | 20 | 100 | 3 | 0.2 | 0 | 4724.6 | 542.8 |
| mixed | PDDM | 20 | 100 | 3 | 0.2 | 0 | -275.6 | 58.9 |
| medium | CLONING | - | - | - | - | - | 4242.4 | 304.5 |
| medium | MBOP | 2 | 100 | 3 | 0.2 | 0 | 5406.5 | 96.6 |
| medium | MBOP-NOPP | 2 | 100 | 3 | 0.2 | 0 | -427.0 | 79.9 |
| medium | MBOP-NOVF | 20 | 100 | 3 | 0.2 | 0 | 4959.8 | 85.5 |
| medium | PDDM | 20 | 100 | 3 | 0.2 | 0 | -331.9 | 30.1 |
| random | CLONING | - | - | - | - | - | -1.0 | 1.1 |
| random | MBOP | 4 | 100 | 3 | 0.8 | 0 | 768.4 | 491.2 |
| random | MBOP-NOPP | 4 | 100 | 3 | 0.8 | 0 | 254.0 | 567.8 |
| random | MBOP-NOVF | 40 | 100 | 3 | 0.8 | 0 | 495.6 | 534.7 |
| random | PDDM | 40 | 100 | 3 | 0.8 | 0 | -156.7 | 110.1 |

Table 6: D4RL HalfCheetah Performance

| Dataset | Policy | Horizon | # Samples | Kappa | Sigma | Beta | Mean | 1-STD |
|---------|--------|---------|-----------|-------|-------|------|------|-------|
| med-expert | CLONING | - | - | - | - | - | 486.6 | 282.4 |
| med-expert | MBOP | 10 | 100.0 | 3 | 0.01 | 0.0 | 1781.7 | 1433.8 |
| med-expert | MBOP-NOPP | 10 | 100.0 | 3 | 0.01 | 0.0 | 151.9 | 30.1 |
| med-expert | MBOP-NOVF | 80 | 100.0 | 3 | 0.01 | 0.0 | 1055.7 | 1300.4 |
| med-expert | PDDM | 80 | 100.0 | 3 | 0.01 | 0.0 | 123.5 | 36.3 |
| medium | CLONING | - | - | - | - | - | 1556.4 | 846.7 |
| medium | MBOP | 4 | 100.0 | 0.3 | 0.01 | 0.0 | 1576.7 | 866.1 |
| medium | MBOP-NOPP | 4 | 100.0 | 0.3 | 0.01 | 0.0 | 124.8 | 65.2 |
| medium | MBOP-NOVF | 40 | 100.0 | 0.3 | 0.01 | 0.0 | 1479.4 | 770.0 |
| medium | PDDM | 40 | 100.0 | 0.3 | 0.01 | 0.0 | 104.7 | 7.8 |
| mixed | CLONING | - | - | - | - | - | 308.2 | 223.2 |
| mixed | MBOP | 4 | 100.0 | 0.3 | 0.02 | 0.0 | 400.5 | 189.1 |
| mixed | MBOP-NOPP | 4 | 100.0 | 0.3 | 0.02 | 0.0 | 141.1 | 46.4 |
| mixed | MBOP-NOVF | 150 | 100.0 | 0.3 | 0.02 | 0.0 | 347.7 | 163.0 |
| mixed | PDDM | 150 | 100.0 | 0.3 | 0.02 | 0.0 | 101.6 | 36.5 |
| random | CLONING | - | - | - | - | - | 289.5 | 6.0 |
| random | MBOP | 4 | 100.0 | 10 | 0.4 | 0.0 | 350.1 | 9.5 |
| random | MBOP-NOPP | 4 | 100.0 | 10 | 0.4 | 0.0 | 81.8 | 42.3 |
| random | MBOP-NOVF | 15 | 100.0 | 10 | 0.4 | 0.0 | 334.4 | 21.1 |
| random | PDDM | 15 | 100.0 | 10 | 0.4 | 0.0 | 44.2 | 12.0 |

Table 7: D4RL Hopper Performance

| Dataset | Policy | Horizon | # Samples | Kappa | Sigma | Beta | Mean | 1-STD |
|---------|--------|---------|-----------|-------|-------|------|------|-------|
| med-expert | CLONING | - | - | - | - | - | 3006.0 | 1844.8 |
| med-expert | MBOP | 2 | 1000 | 1 | 0.05 | 0 | 3222.8 | 1660.7 |
| med-expert | MBOP-NOPP | 2 | 1000 | 1 | 0.05 | 0 | -6.0 | 0.6 |
| med-expert | MBOP-NOVF | 15 | 1000 | 1 | 0.05 | 0 | 2302.7 | 1981.2 |
| med-expert | PDDM | 15 | 1000 | 1 | 0.05 | 0.2 | 209.4 | 113.1 |
| mixed | CLONING | - | - | - | - | - | 528.7 | 335.0 |
| mixed | MBOP | 8 | 1000 | 3 | 0.02 | 0 | 447.1 | 243.8 |
| mixed | MBOP-NOPP | 8 | 1000 | 3 | 0.02 | 0 | 239.3 | 51.5 |
| mixed | MBOP-NOVF | 10 | 1000 | 3 | 0.02 | 0 | 530.0 | 228.8 |
| mixed | PDDM | 10 | 1000 | 3 | 0.02 | 0 | 246.0 | 5.6 |
| medium | CLONING | - | - | - | - | - | 706.8 | 1134.5 |
| medium | MBOP | 2 | 1000 | 0.1 | 0.2 | 0 | 1881.9 | 1350.7 |
| medium | MBOP-NOPP | 2 | 1000 | 0.1 | 0.2 | 0 | -9.9 | 12.8 |
| medium | MBOP-NOVF | 150 | 1000 | 0.1 | 0.2 | 0 | 341.7 | 504.6 |
| medium | PDDM | 150 | 1000 | 0.1 | 0.2 | 0 | -2.7 | 10.3 |
| random | CLONING | - | - | - | - | - | 2.7 | 0.6 |
| random | MBOP | 8 | 1000 | 0.3 | 0.4 | 0 | 371.1 | 252.3 |
| random | MBOP-NOPP | 8 | 1000 | 0.3 | 0.4 | 0 | 484.5 | 268.9 |
| random | MBOP-NOVF | 15 | 1000 | 0.3 | 0.4 | 0 | 220.4 | 124.7 |
| random | PDDM | 15 | 1000 | 0.3 | 0.4 | 0 | 498.9 | 463.0 |

Table 8: D4RL Walker2d Performance

## 5.3 MBOP ABLATIONS

Full results for the various ablations of MBOP are visualized in Figures 4 and 5.

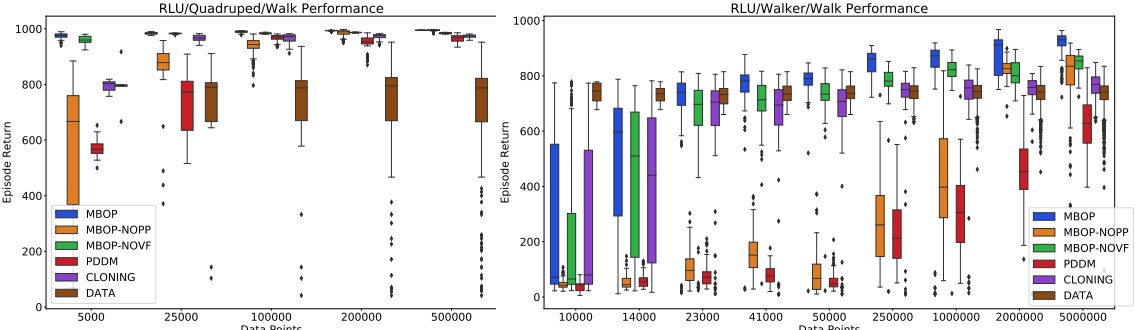

(a) MBOP variants performance on RL-Unplugged RWRL Quadruped Dataset

(b) MBOP variants performance on RL-Unplugged RWRL Walker Dataset

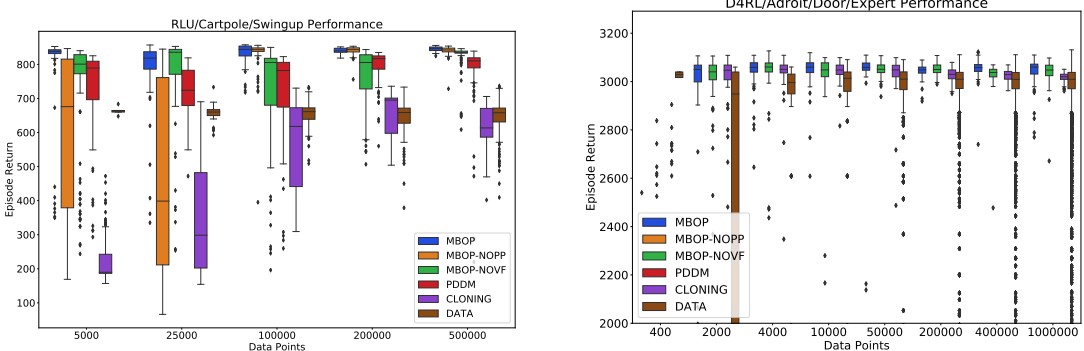

(c) MBOP variants performance on RL-Unplugged RWRL Cartpole Dataset

(d) MBOP variants performance on D4RL - Adroit - door-expert-v0 Dataset

Figure 4: Ablation results on multi-sized datasets form RLU and D4RL.

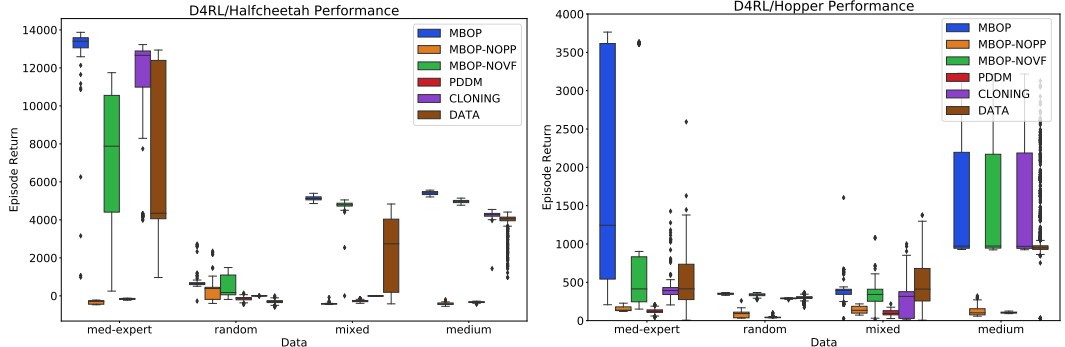

(a) MBOP variants performance on D4RL - HalfCheetah tasks

(b) MBOP variants performance on D4RL - Hopper tasks

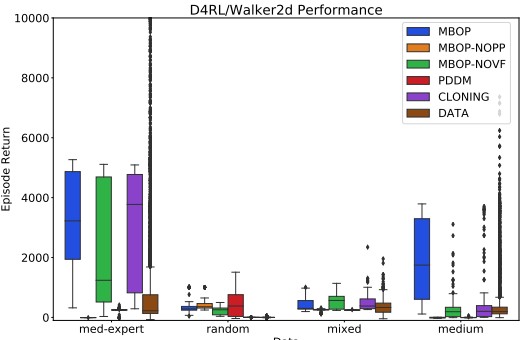

(c) MBOP variants performance on D4RL - Walker tasks

Figure 5: Performance on D4RL tasks from MBOP.

## 5.4 EXECUTION SPEED

| Policy | Horizon | Frequency (Hz) |
|--------|---------|----------------|
| BC | N/A | 362 |
| MBOP | 4 | 106 |
| MBOP | 8 | 71 |
| MBOP | 16 | 40 |

Table 9: `MBOP` maximum control frequencies (steps/second) including simulator time on an Tesla P100 using a single core of a Xeon 2200 MHz equivalent processor.

Execution speeds on the RLU Walker task in represented in Table 9. We see that we can easily achieve control frequencies below 10Hz, but cannot currently attain 100Hz with longer horizons. For lower level control policies for which high-frequency is important, we would suggest distilling the controller into a task-specific policy similar to MoREL (Kidambi et al., 2020) or MOPO (Yu et al., 2020).

## 5.5 MBOP PARAMETERS

All parameters were set as follows except for the D4RL Walker task where we use 15 ensemble networks.

- `# FC Layers:2`
- `Size FC Layers:500`

- `# Ensemble Networks:` 3
- `Learning Rate:` 0.001
- `Batch Size:` 512
- `# Epochs:` 40

### CONTINUED ANALYSIS OF CONSTRAINED TASKS

We can see the height-constrained `Walker` performance in Figure 6a. `MBOP` is able to satisfy the height constraint 80% of the episode while maintaining reasonable performance. Over the various ablations we have found that `MBOP` is better able to maintain base task performance for similar constraint satisfaction rates.

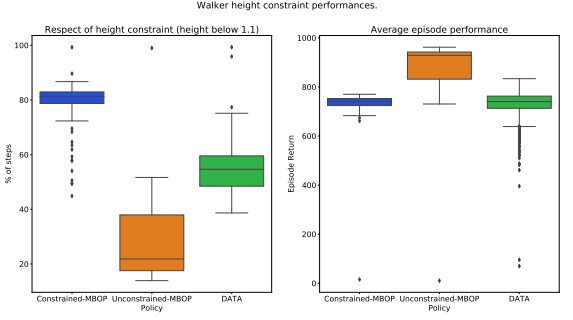

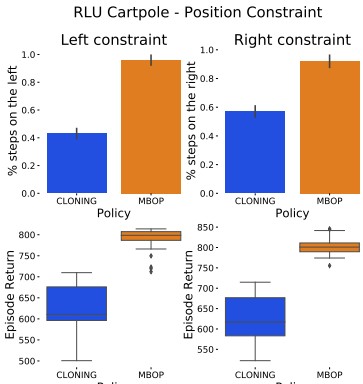

(a) This figure describes the performance of `MBOP` on RLU Walker when constrained to stay below a height threshold. We see that `MBOP` is able to increase the rate of respect of the constraint compared to the behavior policy while maintaining similar episode returns.

(b) Cartpole task constrained to right or left half of the track. We can see that `MBOP` is able to respect the constraint while maintaining performance on both tasks.

Figure 6: Effects of constraints on `MBOP` performance.

### HYPERPARAMETER STABILITY

Figure 7 shows the sensitivity of `MBOP` and associated ablations to the Beta and Horizon parameters. Figure 8 shows the effects of Sigma to `MBOP` and ablations on the RLU datasets. Figure 6b shows sensitivity to Horizon and Kappa in synchrony.

### 5.6 IMPACT OF FILTERING POOR EPISODES

As mentioned in the above part of the paper, for RLU / Quadruped and RLU / Walker we exclude the episodes with lowest returns before training the behavior cloning and value function models. In this section we report the performances on these environment with various filtering thresholds.

For each of these two environments, and each of the dataset sizes we keep a subset of the initial dataset by filtering on the top episodes. We experiments with filters varying from the top-1% to the top-100% (i.e. the entire raw dataset).

| Initial # datapoints | Filtered Top Percent | Mean | 1-STD |
|---|---:|---:|---:|
| 5000 | 1 | 902 | 92 |
| 5000 | 5 | 908 | 64 |
| 5000 | 10 | 897 | 108 |
| 5000 | 20 | 916 | 13 |
| 5000 | 40 | 961 | 30 |
| 5000 | 60 | 951 | 53 |
| 5000 | 80 | 955 | 94 |
| 5000 | 90 | 966 | 12 |
| 5000 | 100 | 960 | 43 |
| 25000 | 1 | 809 | 268 |
| 25000 | 5 | 850 | 229 |
| 25000 | 10 | 973 | 49 |
| 25000 | 20 | 965 | 16 |
| 25000 | 40 | 744 | 329 |
| 25000 | 60 | 365 | 289 |
| 25000 | 80 | 320 | 262 |
| 25000 | 90 | 221 | 178 |
| 25000 | 100 | 115 | 69 |
| 100000 | 1 | 976 | 67 |
| 100000 | 5 | 986 | 4 |
| 100000 | 10 | 987 | 4 |
| 100000 | 20 | 989 | 3 |
| 100000 | 40 | 985 | 23 |
| 100000 | 60 | 876 | 230 |
| 100000 | 80 | 896 | 221 |
| 100000 | 90 | 921 | 177 |
| 100000 | 100 | 547 | 353 |
| 200000 | 1 | 989 | 2 |
| 200000 | 5 | 990 | 2 |
| 200000 | 10 | 993 | 2 |
| 200000 | 20 | 994 | 1 |
| 200000 | 40 | 991 | 2 |
| 200000 | 60 | 990 | 4 |
| 200000 | 80 | 878 | 254 |
| 200000 | 90 | 889 | 259 |
| 200000 | 100 | 876 | 252 |
| 500000 | 1 | 991 | 1 |
| 500000 | 5 | 992 | 1 |
| 500000 | 10 | 995 | 1 |
| 500000 | 20 | 994 | 1 |
| 500000 | 40 | 991 | 2 |
| 500000 | 60 | 992 | 2 |
| 500000 | 80 | 986 | 50 |
| 500000 | 90 | 991 | 3 |
| 500000 | 100 | 991 | 2 |

Table 10: MBOP performance on RLU / Quadruped with various filtering thresholds for top episodes

| Initial # datapoints | Filtered Top Percent | Mean | 1-STD |
|---|---|---|---|
| 50000 | 1 | 49 | 63 |
| 50000 | 5 | 86 | 112 |
| 50000 | 10 | 195 | 252 |
| 50000 | 20 | 243 | 293 |
| 50000 | 40 | 636 | 269 |
| 50000 | 60 | 750 | 182 |
| 50000 | 80 | 772 | 119 |
| 50000 | 90 | 719 | 201 |
| 50000 | 100 | 770 | 124 |
| 250000 | 1 | 111 | 113 |
| 250000 | 5 | 397 | 397 |
| 250000 | 10 | 410 | 406 |
| 250000 | 20 | 810 | 171 |
| 250000 | 40 | 848 | 42 |
| 250000 | 60 | 842 | 45 |
| 250000 | 80 | 836 | 46 |
| 250000 | 90 | 848 | 45 |
| 250000 | 100 | 838 | 43 |
| 1000000 | 1 | 154 | 201 |
| 1000000 | 5 | 670 | 348 |
| 1000000 | 10 | 870 | 88 |
| 1000000 | 20 | 858 | 101 |
| 1000000 | 40 | 858 | 63 |
| 1000000 | 60 | 859 | 97 |
| 1000000 | 80 | 851 | 47 |
| 1000000 | 90 | 847 | 53 |
| 1000000 | 100 | 855 | 46 |
| 2000000 | 1 | 618 | 386 |
| 2000000 | 5 | 741 | 348 |
| 2000000 | 10 | 859 | 194 |
| 2000000 | 20 | 876 | 111 |
| 2000000 | 40 | 867 | 62 |
| 2000000 | 60 | 860 | 59 |
| 2000000 | 80 | 888 | 55 |
| 2000000 | 90 | 873 | 60 |
| 2000000 | 100 | 858 | 61 |
| 5000000 | 1 | 639 | 404 |
| 5000000 | 5 | 875 | 179 |
| 5000000 | 10 | 909 | 37 |
| 5000000 | 20 | 907 | 49 |
| 5000000 | 40 | 892 | 60 |
| 5000000 | 60 | 892 | 58 |
| 5000000 | 80 | 853 | 54 |
| 5000000 | 90 | 875 | 63 |
| 5000000 | 100 | 863 | 65 |

Table 11: MBOP performance on RLU / Walker with various filtering thresholds for top episodes

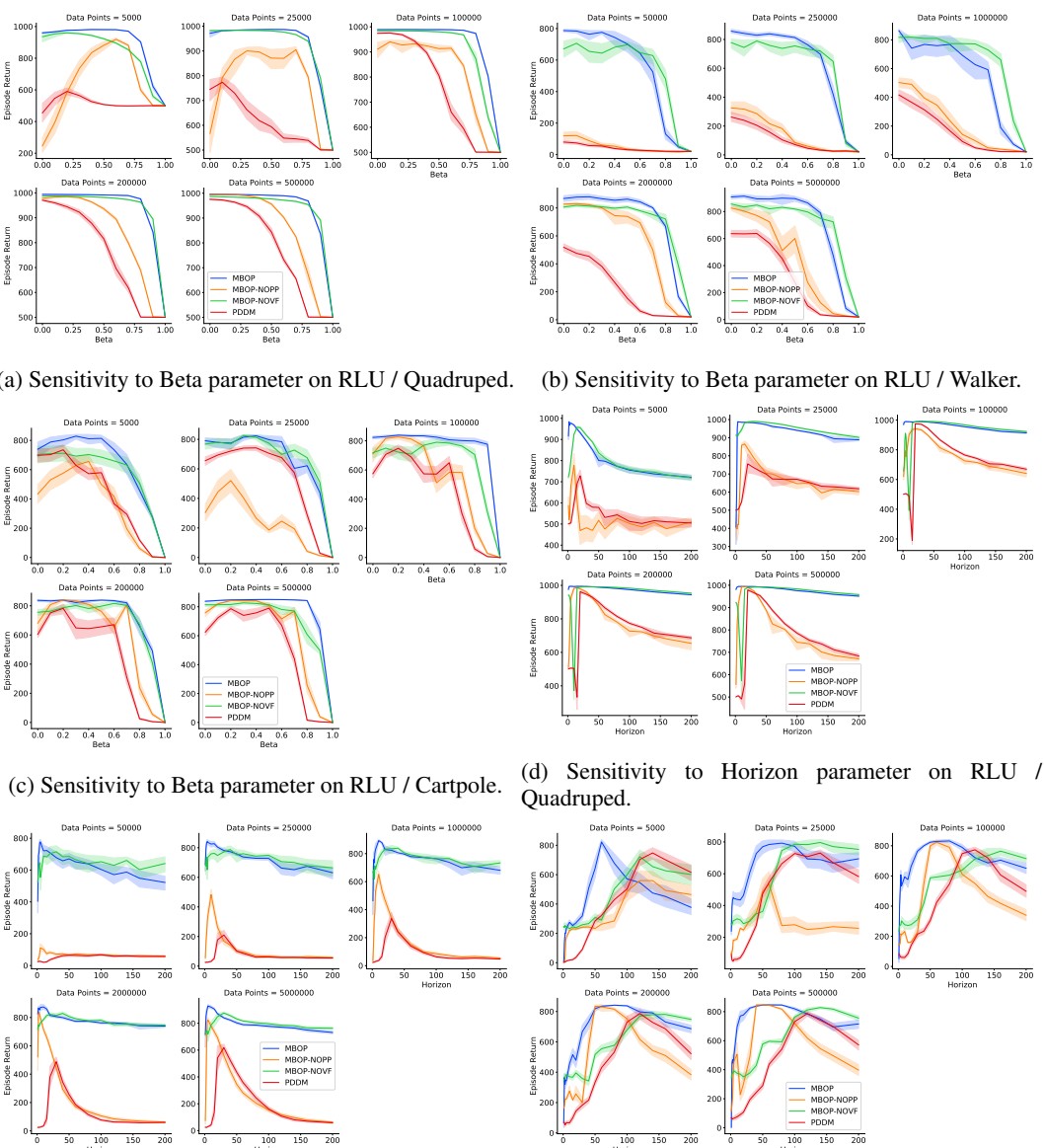

(a) Sensitivity to Beta parameter on RLU / Quadruped. (b) Sensitivity to Beta parameter on RLU / Walker.

(c) Sensitivity to Beta parameter on RLU / Cartpole. (d) Sensitivity to Horizon parameter on RLU / Quadruped.

(e) Sensitivity to Horizon parameter on RLU / Walker. (f) Sensitivity to Horizon parameter on RLU / Cartpole.

Figure 7: MBOP sensitivity to Beta & Horizon on RLU datasets.

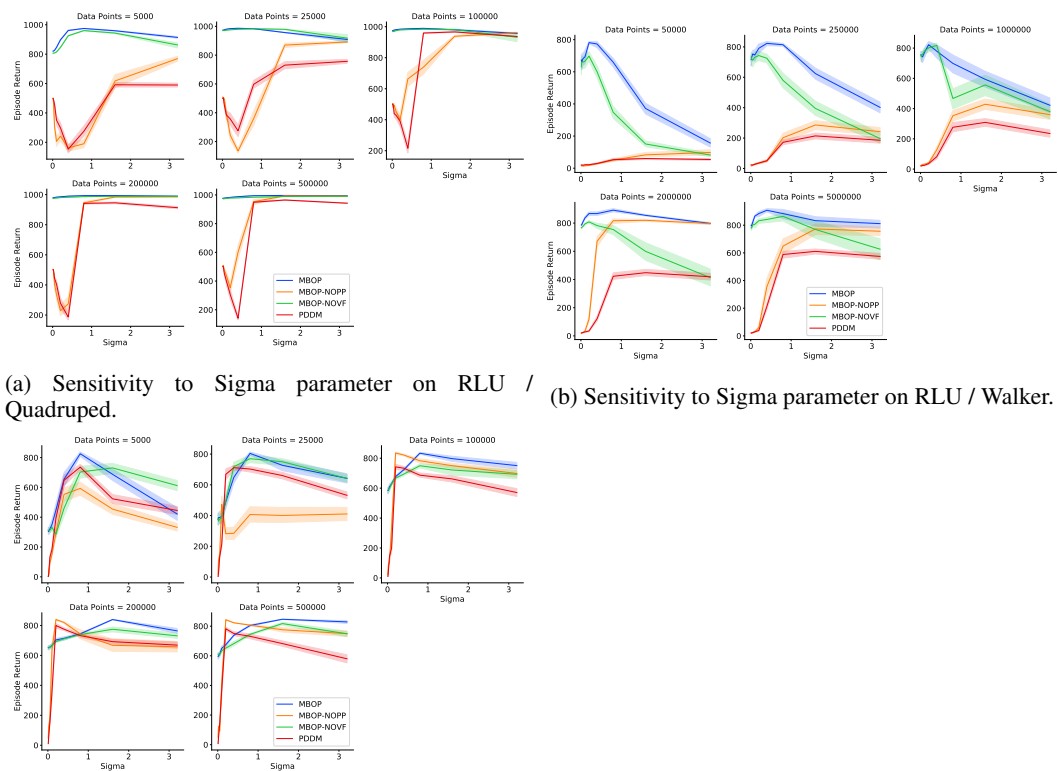

(a) Sensitivity to Sigma parameter on RLU / Quadruped.

(b) Sensitivity to Sigma parameter on RLU / Walker.

(c) Sensitivity to Sigma parameter on RLU / Cartpole.

Figure 8: MBOP sensitivity to Sigma on RLU datasets.

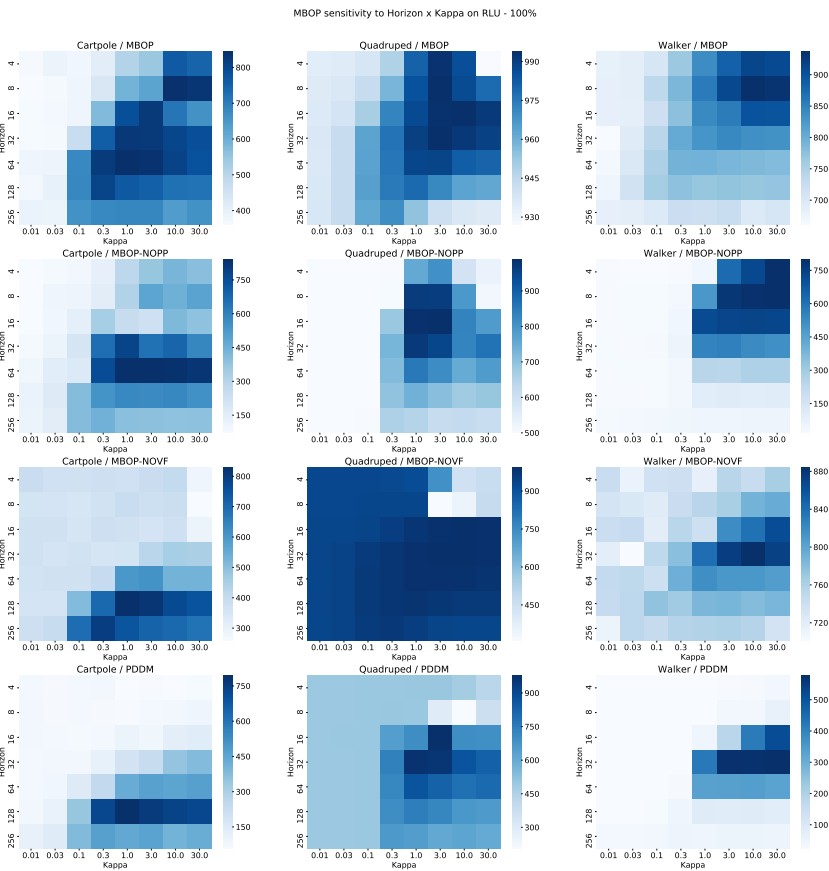

Figure 9: Sensitivity to Horizon x Kappa on RLU environments (full datasets). Legend represents average episode return.

