# OpenReview forum: "Model-Based Offline Planning"
_ICLR.cc/2021/Conference — ICLR 2021 Poster_

### Official Review · AnonReviewer1 · 2020-10-18
**Interesting approach for planning in offline RL - allowing for small data regimes and good transfer**

**Rating:** 7
**Confidence:** 4

**Review:**

This paper leverages model predictive control (MPI) and proposes an approach which utilizes a transition+reward model, expected return model, and behavior-cloned policy to plan in a fully offline setting. The authors conduct experiments and ablation studies, showing the benefit of their approach in small data regimes and transferability to secondary objectives (e.g., constraints).

Positive feedback:

1. MBOP is an interesting approach to utilizing environmental model together with a behaviour-cloned policy in an MPI framework. Its main novelties w.r.t PDDM include the use of the behavior-cloned policy and an expected return model.
2. MBOP shows to be a strong baseline for offline reinforcement learning, especially in small data regimes, which is especially interesting. Their ablation studies show the importance of using behavior cloning.
3. Unlike other methods for offline-RL which first learn models and then use them in to solve the underlying MDP, MDOP in its essence allows to learn secondary objectives without the need to retrain the policy on the new MDP. The constrained objectives are especially interesting with this respect.

Questions and concerns:

1. It was unclear (even after reading the information in the appendix) how the models f_m, f_R and f_b are trained (e.g., when and if ensembles are used, etc.)

2. The method is fundamentally different than MOPO and MBPO, and it not clear if the reason for improvements are due to fundamental algorithmic factors. Some examples:
- The authors use the same model head for simulating the full trajectory.
- The use the mean reward and not the maximum reward as in MOPO. MBOP also learns an expected return model in addition to the reward model in MOPO.
- Are the same transition models used in MBOP as in MOPO and MBPO?
Such factors may greatly affect performance and overall results. If these are the factors that affected the boost in performance, it would greatly reduce the general interest of the proposed approach. It is thus currently unclear if the comparison is fair.

3. The authors chose to discard episodes that are below a certain threshold for the training of f_b and f_R. This raises a question as to the ability of the algorithm to utilize non-optimal data, e.g., dirty data that wasn’t collected by a quasi-optimal policy, or a mixture of policies.

4. As MBOP utilizes behavioral cloning at its core, it would be interesting to see how it compares to behaviour regularized offline reinforcement learning: https://arxiv.org/pdf/1911.11361.pdf

5. While I understand this work is a generalization of previously developed methods (MPPI, PDDM), some of its underlying assumptions are unclear. In MBOP actions are linearly interpolated (beta mixtures and weighted trajectories). When and why do the authors assume such interpolation should work?

6. There are two flaws that make the applicability of MBOP somewhat problematic:

- Lots of hyper parameters - unclear how these should be chosen. This includes the many different models which are not fully discussed.

- Uncertainty of the algorithm is not taken into account by MBOP. I believe this to be an essential requirement for an offline-RL algorithm to be applicable, as no new experience can be collected.

7. Finally, the clarity of the paper could greatly be improved. Overall it feels like the paper is not self contained and hard to understand. Many notations are not explained, and some of the taken approaches are unclear. I feel that Algorithm 2 should be explained better. The authors should provide more detailed information on what each line signifies and why it is an important feature of their approach.
Minor question: What does "min" signify in T_{min} in the Algorithm 2, line 11?



Typos:

Section 2.4:

- “…is a truncated value function, which *provides* the expected”

Section 4.3:

- “*One* of the main…”
- “We can *then* adapt the trajectory…”

---

> ### Author Response · Authors · 2020-11-24
> **Response to AnonReviewer1 (1/3)**
>
> *It was unclear (even after reading the information in the appendix) how the models f_m, f_R and f_b are trained (e.g., when and if ensembles are used, etc.)*
>
> **The models are indeed ensembles of feed-forward neural networks.  Each ensemble member is initialized differently and trained with a different order of mini-batches.  Other than this, the training regime is as one would expect for supervised deep ML model training (ADAM, fixed LR, early stopping).  We will add a section that clarifies this.**
>
> *The method is fundamentally different than MOPO and MBPO, and it not clear if the reason for improvements are due to fundamental algorithmic factors. Some examples:*
> * *The authors use the same model head for simulating the full trajectory.*
>
> **We use several model heads independently for each trajectory and use the same head within a trajectory, in order to have consistency within a trajectory but variations across trajectories.  This is a design decision used in both PDDM and other prior work and we decided to stick with it as discussions with PDDM authors suggests it works better than round-robin between model heads.**
>
> * *They use the mean reward and not the maximum reward as in MOPO. MBOP also learns an expected return model in addition to the reward model in MOPO.*
>
> **If we understand correctly you are referring to the averaging over the ensemble heads.  This is a design decision taken from PDDM, and is a direction we intend to look at in future work.
> Empirically, taking the maximum seems to work well and as it had been used previously in the literature it seemed reasonable to continue doing so in the current iteration of our work.
> We can add a clarification in the paper.**
>
> * *Are the same transition models used in MBOP as in MOPO and MBPO? Such factors may greatly affect performance and overall results. If these are the factors that affected the boost in performance, it would greatly reduce the general interest of the proposed approach. It is thus currently unclear if the comparison is fair.*
>
> **First of all, although I believe this is clear to the reviewer, I want to reiterate that the MBOP and MOPO/Morel algorithms are different in the following way:**
> * **MBOP leverages online planning. This provides the ability to change the task online without retraining.**
> * **MBOP and MOPO perform well on different data regimes which shows fundamental differences between these algorithms (and the benefit of MBOP on some data regimes)**
>
> **At a high level the transition models in MOPO/MBPO and MBOP are indeed the same (bootstrap ensemble of feed-forward neural nets) which are then used to generate possible future states.  Therefore it suggests the general approach is more linked to performance gains rather than underlying learning machine techniques.  However, we want to point out that we don’t believe raw performance is the main interest of mbop, but rather its good data efficiency and ability to deal with zero-shot adaptation or other dynamic goals and constraints, which can be quite useful in a real system.  The fact that it has these nice features and also performs reasonably or even better than similar methods is somewhat of a ‘bonus’ in our eyes (we are not intending to chase SoTA).**

---

> > ### Author Response · Authors · 2020-11-24
> > **Response to AnonReviewer1 (2/3)**
> >
> > *The authors chose to discard episodes that are below a certain threshold for the training of f_b and f_R. This raises a question as to the ability of the algorithm to utilize non-optimal data, e.g., dirty data that wasn’t collected by a quasi-optimal policy, or a mixture of policies.
> > Agreed, this is a performance decision.*
> >
> > **We can maintain the data’s presence and performance is somewhat reduced, we did not fully analyze the extent of the effects but from memory performance was reduced by approx. 5%.  The inclusion of the mixed source datasets in the d4rl environments is intended to demonstrate less uniform datasets.  Although pruning does ‘clean up’ the data, it is also a tradeoff because it also reduces the amount of data to be used but in the case of MBOP this compromise seems to make sense at least on these tasks.**
> >
> > **MBOP models are trained like standard supervised ML models. Like for many supervised ML problems, there is a tradeoff between data quality and data quantity. Indeed the more data the model is trained on the better it tends to generalize however removing outliers and noisy records - especially for smaller datasets - can help with model performance.**
> >
> > **This quantity / quality tradeoff is also relevant to MBOP models.
> > Regarding the environment model, we always use all the data available - the reasoning being that all the transitions available have value for learning.
> > We only analyze filtering out poor performing episodes for the behavior cloning model and for the value function.
> > Among all the 7 environments considered we only applied the filtering method to 2 of them (RLU/Walker and RLU/Quadruped).
> > The improvement due to episode filtering is minor for most dataset and we believe in most cases our algorithm is still valuable without such an episode filtering capability.  We have mentioned this in the paper and have added unfiltered data results for these two tasks to the appendix.**
> >
> > **To your point on the ability of our algorithm on non-optimal data. Even though on lower quality dataset we do not match SoTA approaches like MOPO and MoREL, our algorithm still outperforms all the times our PDDM baseline, hence reinforcing our paper contribution that introducing a BC and VF to PDDM increases performance for all data regimes.
> > We believe that this is still an interesting contribution - MOPO and MoREL being a different class of algorithm all together.**
> >
> > **We also mention directions for future work to further extend MBOP to more noisy datasets using a stochastic prior.**
> >
> >
> >
> > *As MBOP utilizes behavioral cloning at its core, it would be interesting to see how it compares to behaviour regularized offline reinforcement learning: https://arxiv.org/pdf/1911.11361.pdf*
> >
> > **We did not compare directly to BRAC as it is a model-free method and does not have many of the aspects that MBPO brings to the table.  If  we compared to BRAC it also opened the door to comparing to a whole slew of other baselines which seemed to start leaving the scope, once again our argument is not so much that MBOP is a SoTA method (although it seems to present good performance scores), rather that it is a novel approach to offline policy learning that includes the niceties of zero-shot adaptation and lighter training regimes.  Nevertheless when looking at BRAC's performance on D4RL (Table 1 of D4RL paper) we can see that MBOP is similar in performance on medium datasets, and although worse on random and medium data, it is generally more robust to failure (scores of 0) compared to similar algorithms.  We will try to address this in the paper but fear lack of space will constrain us.**
> >
> > *While I understand this work is a generalization of previously developed methods (MPPI, PDDM), some of its underlying assumptions are unclear. In MBOP actions are linearly interpolated (beta mixtures and weighted trajectories). When and why do the authors assume such interpolation should work?*
> >
> > **The authors agree that the use of MPPI restricts the controller to somewhat stable and unimodal control laws.  We agree with this observation, and also agree these assumptions are a bit strong.  We hope to look at removing these assumptions in follow-up work but believe the observation that these assumptions are empirically functional in many simulated tasks to nevertheless be a worthwhile point to share with the community.**

---

> > > ### Author Response · Authors · 2020-11-24
> > > **Response to AnonReviewer1 (3/3)**
> > >
> > > *There are two flaws that make the applicability of MBOP somewhat problematic:*
> > > *Lots of hyper parameters - unclear how these should be chosen. This includes the many different models which are not fully discussed.*
> > >
> > > **The authors don't believe there are more hyperparameters in mbop than in any SoTA Deep RL method.  Additionally, many of these parameters are specific to supervised training of each model and can be evaluated offline (e.g. number of epochs, batch size, learning rate, etc.). A lot of these are actually very stable and we used the same value across all of our experiments.**
> > > **MBOP has specifically 5 hyper parameters: horizon, sample size, kappa (for return weighting), beta (for trajectory consistency) and sigma for noise.
> > > 1/ This is the exact same number of parameters as PDDM - i.e. MBOP does not introduce any new hyper parameter to PDDM - which has shown its value.
> > > 2/ We performed a stability analysis with respect to variation of these parameters and show that:
> > > Within one environment and dataset there is stability with respect to hyper-parameter variation
> > > Across environments and dataset, the top performances are reached approximately for the same hyper-parameter values.**
> > >
> > > *Uncertainty of the algorithm is not taken into account by MBOP. I believe this to be an essential requirement for an offline-RL algorithm to be applicable, as no new experience can be collected.*
> > >
> > > **Uncertainty is not explicitly materialized as a quantity in the control algorithm, but it is implicitly leveraged through the sampling of ensemble heads during trajectory rollouts in MPPI.  This is the current SoTA approach for leveraging ensembles in MPC-based methods as far as the authors know, but we also agree that there is more to be done here.  As per our point above, this is a clear point we would like to look at improving, especially if we want to then bring the controller on-line and continue exploring.  However, the current approach performs almost optimally on many of the benchmark tasks so we still believe this is an interesting algorithm to share with the community as work continues in leveraging these aspects more explicitly.**
> > >
> > > *Finally, the clarity of the paper could greatly be improved. Overall it feels like the paper is not self contained and hard to understand. Many notations are not explained, and some of the taken approaches are unclear. I feel that Algorithm 2 should be explained better. The authors should provide more detailed information on what each line signifies and why it is an important feature of their approach. Minor question: What does "min" signify in T_{min} in the Algorithm 2, line 11?*
> > >
> > > **We are sorry to hear that the paper was hard to follow.**
> > >
> > > **Regarding `T_{min}` the paper actually mentions `T_{min(t, H-1)}`, this is directly the mathematical `min` function that takes the smallest value of the set {t, H-1}. This means that if we sample beyond the trajectory horizon we re-use the latest step of the trajectory.
> > > At each iteration we re-use the previous sampled trajectory of length H. We discard the first step of the trajectory which leads to a trajectory of length H-1 that is used to generate the new trajectory. To keep a trajectory of length H, we duplicate the last step - hence the min(t, H-1).**
> > >
> > > **We will add some comments to Alg. 2 and try to provide a text-based complement that should hopefully make it easier to follow.**

---

> > > > ### Comment · AnonReviewer1 · 2020-11-25
> > > > **Thank you for your response**
> > > >
> > > > The reubttal answered most of my questions and the authors' response did answer some of my concerns. Though I feel that the overall algorithmic approach can still be improved, I do not find this as a reason to reject the paper. I have increased my score to 7 accordingly.

---

> ### Author Response · Authors · 2020-11-24
> **Response to AnonReviewer1 (Intro)**
>
> Hello and thank your for your thorough review of the MBOP paper.  We appreciate the time taken and your thorough comments.  I have responded below to specific points.  More generally we will try to increase the clarity in the paper for which we are just finishing up the revision right now.    Overall, I just want to point out that the MBOP approach provides a new class of policies for offline problems, and although similar approaches exist in the online space, we show through our ablations that they did not adapt out of the box to the offline scenario.  I think this is the main gist of the paper, and performance comparisons are primarily to show that we achieve the same class of performance, but we don't particularly seek to beat SoTA, as the added ability of zero-shot adaptation is, in our eyes, a very practical thing to have when running on a real system.  We will try to make our contribution more clear, in addition to responding to other reviewers' concerns, but hope that in the following responses to your questions we also clear up some of your concerns.  Hopefully this can help show the work in a clearer light and convince both yourself and fellow reviewers to consider this work for acceptance at ICLR 2021.
>
> Best,
>
> The MBOP Authors

---

### Official Review · AnonReviewer2 · 2020-10-28
**Review of MBOP**

**Rating:** 5
**Confidence:** 5

**Review:**

**Summary**

This work studies the offline RL problem and proposes MBOP for the same. The proposed method learns ensembles of dynamics models, behavioral policies, and value functions using the offline dataset. Subsequently, the approach uses online MPC with a learned terminal value function. The paper demonstrates experimental results on standard benchmark tasks (RLU and D4RL) as well as zero-shot adaptation results.

The strengths of the paper are showing results with an MPC-based approach and zero-shot adaptation results. My main concerns with the paper are exaggerated novelty, improper attribution and missing related works, and narrowly applicable approach. In the current form/version, I do not feel comfortable recommending acceptance. However, I am open to considering again if the authors can address the concerns during the rebuttal stage (which I expect will require a major revision).

----

(1) **Exaggerated novelty:** This paper must be considered in the context of a number of recent model-based offline RL papers. In particular, after MOReL and MOPO (which have been on arXiv for ~5 months before ICLR deadline), the use of model-based methods for offline RL, by itself, can no longer be considered a novelty. What sets this apart from MOReL and MOPO is the use of MPC vs a dyna style policy learning algorithm. This distinction and difference is not surprising or new, since a number of papers have explored this theme in the standard RL setting. However, these papers have not be adequately cited or discussed (e.g. guided policy search, POLO [1], AOP [2] etc). The MBRL tutorial (https://sites.google.com/view/mbrl-tutorial) has additional references too. I would recommend the authors rewrite the paper, to better situate the contributions of this work relative to what is known in literature. Right now, the first mentions of the most relevant prior work (MOReL and MOPO) appear only in page 5.

(2) **Missing related work:** Pointers to relevant works are missing. Notable omissions include Dyna based MBRL algorithms (e.g. Game-MBRL [3] and MBPO [4]). This also ties with the previous point about exaggerated claims of novelty. Furthermore, Game-MRBL also explores the adaptability of MBRL in detail. When the dynamics is the same but the reward function changes, then the approach considered in MBOP will work. However, when the dynamics is inconsistent or non-stationary (e.g. data for locomotion collected on the road but robot enters dirt on sidewalk), then the MBOP approach would fail rapidly.

The planning algorithm in PDDM (filtering + reward refinement) is based on the MPC algorithm used in POLO [1]. I would recommend the authors to cite both papers for proper scientific dissemination of information. Line 11 in algorithm 2 can be interpreted as a "learning rate" to shift a_t from current value towards a target value. This was proposed and analyzed in the work of Wagener et al. [4]. I recommend the authors to cite and discuss this work. The Adroit tasks were proposed in [5] but this paper is not cited.

Line 6 in Alg2 is equivalent to running a for loop over the models in the ensemble and an inner for loop running for N/K trajectories per model. Can the authors clarify if this is true? If so, the consistent trajectory sampling approach has been utilized in a number of prior works including Game-MBRL.

(3) **Narrowly applicable solution approach** : The paper relies on the use of a behavior cloning policy for regularization and assumes a single consistent policy is used to obtain the offline dataset. If multiple policies are used to construct the dataset, will the MBOP approach work? Do you propose to learn multi-modal generative models for the policies? Do you have experimental results in this regime?

----

**Update after author response**

I appreciate the authors efforts in the revision. This has addressed my concerns about the related work and (partly) exaggerated novelty. As a result I have increased the score from a 4 to 5.

In my opinion, the distinction between MPC and policy optimization is minor -- policy optimization when given enough iterations can match an optimal planner, and similarly MPC when equipped with a good terminal value function can match a policy learner. The practical differences between MPC and policy learning have also been extensively explored in prior work (e.g. GPS, POLO, AOP etc). Thus, while MBOP provides an interesting case study in the use of MPC for offline RL, there is limited novel insight. Nevertheless, I appreciate the authors for the thoroughness of the case studies, and for updating the paper based on reviewer feedback.

---

**References**

[1] Lowrey et al. Plan Online Learn Offline. ICLR 2019.

[2] Lu et al. Adaptive Online Planning for Continual Lifelong Learning. 2019.

[3] Rajeswaran et al. A Game Theoretic Framework for Model Based Reinforcement Learning. ICML 2020.

[4] Janner et al. When to Trust Your Model: Model-Based Policy Optimization. NeurIPS 2019.

[5] Wagener et al. An Online Learning Approach to Model Predictive Control. RSS 2019.

[6] Rajeswaran et al. Learning Complex Dexterous Manipulation with Deep Reinforcement Learning and Demonstrations. RSS 2018.

---

> ### Author Response · Authors · 2020-11-24
> **Final response to AnonReviewer2**
>
> Hello,
>
> As am I finishing the revised version we have some more specific responses to your review and to the AC discussion:
>
> Overall: We have re-structured the paper, placing the related works in Section 2 (just after the intro), and adding a clear paragraph in the intro (2nd paragraph) that describes existing approaches to offline RL, and clearly situates MBOP relative to MoREL and MOPO.  We have also added additional missing related works to the related works section, tried to make the specific contribution of MBOP clear, and generally make clear that we don't believe MBOP's novelty goes beyond the stated claims and that there exists a large baggage of existing work that already introduces many of the ideas used in MBOP, just never in the way or for the use that MBOP has.
>
> 1) Exaggerated novelty:  We have re-structured the paper to make contributions clearer, and quickly contextualize existing works in this space from the 2nd paragraph of the introduction.  Hopefully this will clarify our contribution relative to existing work and not give the impression of overstating anything.
>
> 2) We admittedly missed a couple key citations and want to thank the reviewer for bringing these up.  We have added AOP and Game-MBRL as related works dealing with zero-shot adaptation.  MBPO is actually one of our baselines, although the citation was broken.  We have since mentioned it in related works in relation to MoREL and MOPO (and have added Dreamer).   POLO is clearly discussed in the related works (Sec. 2 Paragraph 3). We have added GPS as a foundational citation and AOP has been mentioned in related works on task adaptation.  We have also added the proper citations for OpenAI Gym, DeepMind Control Suite and the Adroit tasks.  Line 11 of Alg. 2 describes PDDM, which is contemperaneous with Wagener et al.  We have added a reference to this work as well.  Line 6 Alg 2 is taken directly from the PDDM codebase on github, so we considered it part of our clearly cited dependance on PDDM.
>
> 3) The current approach indeed assumes the behavior policy generating the data is somewhat reasonable and unimodal.  This is a reasonable assumption in many real-world systems with a human operator where one would want to learn to replicate the operator's commands and potentially improve on them.  Follow-up work will look at using multi-modal priors and better supporting strongly sub-optimal behavior policies.  We have touched upon this in the conclusion.
>
> Hopefully this restructuring and editing work will be sufficient to satisfy your concerns on exaggerated novelty and missing related work.  Regarding the narrowly applicable solution concern, many current offline RL approaches require a somewhat decent behavior policy, either due to constraining their behavior to the behavior policy's, or even just needing MDP coverage around the solution space.  We believe many real-world tasks could take advantage of an approach such as MBOP's and hope to generalize our approach further in upcoming work.

---

### Official Review · AnonReviewer3 · 2020-10-30
**Have potential aplications, but needs improvement**

**Rating:** 5
**Confidence:** 3

**Review:**

This paper offered a model-based offline planning by taking advantage of learned behavior-cloning  policy and learned value function in addition to learning the model of the environment. Model learning is adapted from a work from Nagbandi et al. As the method takes advantage of behavior-cloning  policy, its performance boosts when the prior policy is reasonably well (medium or expert level). Moreover, learning value function helps the method to achieve higher rewards compared to the behavior-cloning  policy. Having said that, at least from the algorithmic point of view, this work does not seem to have a huge contribution. Moreover, it fails to provide a good solution compared to other model based learning methods when the behavior-cloning policy is not good (although it performs better than the BC policy). In fact, it looks like it would be a suitable method only when behavior-cloning  policy is in medium level (For the expert level vanilla BC gives comparable result). While this could have potential applications, the current work does not offer examples of it. If the authors agree to my conclusion about the medium level BC, finding (or even mentioning) the possible scenarios and environments could improve the impact of their work (And if they don't agree I am eager to know why). Moreover, it looks like due to their model-learning algorithm, the application of the method is mainly limited to robotics.
Another point I want to raise is the conceptual similarity of this work with Value Iteration Networks (VINs). I think VINs are mainly for discrete state space tasks, but I am curious to know authors' opinion on this. Moreover, I think it is useful to mention it in the paper as a related work.
Also, the authors mentioned that  "Along with this high-level design, many implementation details such as consistent ensemble sampling during rollouts, or averaging returns over ensemble heads, appear to be important for a stable controller from our  experience." which I believe is in favor of their potential contribution. However, in the conclusion they  said "MBOP is an easy to implement [...] algorithm". These two seem contradicting to me.

Update:
I thank the authors for their response and increase my score by 1.

---

### Official Review · AnonReviewer4 · 2020-10-30
**Good contribution to model based Offline RL. Well written, thorough evaluation**

**Rating:** 8
**Confidence:** 4

**Review:**

##########################################################################

Summary:

The paper proposes an algorithm to improve on a 'semi-performant' policy on a real world system from logged data of this policy driving the system (offline setting).
The algorithm build a MPC based controller using as ingredients learnt models of the dynamics, reward and value function along with a clone of the semi performant policy.

The algorithm mixes in an original fashion the following ingredients
- soft reward weighted trajectory averaging to select the next action from sampled trajectories (PDDM).
- reuse of past MPC trajectories (linear mixture with next guiding samples)
- guided shooting with a cloned policy (POPLIN)
- learning of ensembles to capture a notion of learnt model uncertainty (PETS)


##########################################################################

Reasons for score:

The combination of the aforementioned ideas to the problem of offline policy improvement is original.
The paper is well written, the related work section is thorough the results  convincing.


##########################################################################
Remarks:

1. The model based approach to off-line learning is a very promising avenue. The explicit separate representation of reward and dynamics provides great flexibility as shown in the adaptation to new objectives or additional constraints.

2. The paper is close to PDDM as acknowledged by the authors in the introduction sections.
The ablation study is interesting and shows that the behavior cloning is the key factor in the algorithm.
The reader is referred to Figure 2 and the results not discussed at all.
I guess it is a space issue, but I'd like to read it.

3. In the ablation section, the author say they recover PDDM when removing the policy and the value function.
I would suggest you give a more central place to this statement.
The RL literature is (incredibly) messy and discussing relation (inclusion, generalization, special case) between methods
when available should be highlighted more.

#########################################################################

Some typos:

(1) sec 2.1 'aim to to'

---

> ### Author Response · Authors · 2020-11-24
> **Response to AnonReviewer4**
>
> Hello, and thank you for your time spent reviewing as well as your kind appreciation of our contribution.  I find the summary and reasoning for the score to be a good summary of our contributions.  Especially Remark 1 on the possible flexibility of the proposed approach is what we hold close to heart with this approach, especially for real systems for which the offline aspect can be quite a boon.
>
> We will add some discussion on Figure 2 and elaborate on the ablation study, indeed space was a bit tight but we now have an extra page.    We will also try to make the recovery of PDDM more clear to better situate the contribution in existing algorithm families and fix the mentioned typo.
>
> Best,
>
> MBOP Authors

---

### Author Response · Authors · 2020-11-25
**Generel Response**

We want to begin by thanking all reviewers for their time. We have significantly restructured the paper to put forward the related works and better contextualize and clarify our contribution.  We have also clarified Algorithm 2 with additional text, clarification on notations, and comments in the algorithm.  We have also tried to make more clear in the introduction, conclusion, and algorithmic section the possible use cases of MBOP, even if there are clearly data regimes where it is not a great solution (random or highly multi-modal data).  We believe these changes make this a stronger and more clear scientific contribution, and hope certain reviewers will reconsider their position in light of these modifications and answers to their questions.  Unfortunately because there are so many changes to the paper, it was not practical to highlight them with a different color, so we have tried to point out in the reviewer comments where the changes may have appeared.

All the best,

MBOP Authors

---

### Decision · Program_Chairs · 2021-01-07
**Final Decision**

**Decision:**

Accept (Poster)

**Comment:**

# Quality:
The experimental evaluation is thorough and well designed.

# Clarity:
After rebuttal, the paper is well written and clear in its contributions.

# Originality:
The proposed approach builds over existing literature, as clearly indicated in the manuscript and acknowledged by the authors in the rebuttal, by mixing existing contributions in a novel fashion.

# Significance of this work:
This work deal with an important and timely topic. The experimental results are convincing and demonstrate strong performance for the proposed approach.

# Overall:
This manuscript provides an incremental but solid contribution to the topic of model-based reinforcement learning.

# Personal comments:
- I disagree with Reviewer2, in that I feel that after the rebuttal the manuscript clearly and thoroughly states prior works and the novelty of the proposed approach. This is not a survey paper and should be hold to realistic standards.